# Endocannabinoid signaling regulates the reinforcing and psychostimulant effects of ketamine in mice

Wei Xu[1,3], Hongchun Li[1,3], Liang Wang[1], Jiamei Zhang[1], Chunqi Liu[1], Xuemei Wan[1], Xiaochong Liu[1], Yiming Hu[1], Qiyao Fang[1], Yuanyuan Xiao[1], Qian Bu[1], Hongbo Wang[2], Jingwei Tian[2], Yinglan Zhao[1] & Xiaobo Cen [1✉]

The abuse potential of ketamine limits its clinical application, but the precise mechanism remains largely unclear. Here we discovered that ketamine significantly remodels the endocannabinoid-related lipidome and activates 2-arachidonoylglycerol (2-AG) signaling in the dorsal striatum (caudate nucleus and putamen, CPu) of mice. Elevated 2-AG in the CPu is essential for the psychostimulant and reinforcing effects of ketamine, whereas blockade of the cannabinoid CB1 receptor, a predominant 2-AG receptor, attenuates ketamine-induced remodeling of neuronal dendrite structure and neurobehaviors. Ketamine represses the transcription of the monoacylglycerol lipase (MAGL) gene by promoting the expression of PRDM5, a negative transcription factor of the MAGL gene, leading to increased 2-AG production. Genetic overexpression of MAGL or silencing of PRDM5 expression in the CPu robustly reduces 2-AG production and ketamine effects. Collectively, endocannabinoid signaling plays a critical role in mediating the psychostimulant and reinforcing properties of ketamine.

[1] National Chengdu Center for Safety Evaluation of Drugs, State Key Laboratory of Biotherapy/Collaborative Innovation Center for Biotherapy, West China Hospital, Sichuan University, 610041 Chengdu, People's Republic of China. [2] Ministry of Education, Collaborative Innovation Center of Advanced Drug Delivery System and Biotech Drugs in Universities of Shandong, Yantai University, 264005 Yantai, People's Republic of China. [3]These authors contributed equally: Wei Xu, Hongchun Li. ✉email: xbcen@scu.edu.cn

Ketamine, a noncompetitive antagonist of the N-methyl-D-aspartate (NMDA) receptor, has been widely used as a dissociative anesthetic agent, effective analgesic, procedural sedative, and refractory antidepressant[1]. However, the reinforcing and rewarding properties of ketamine make it a substance with abuse potential and limit its clinical use. The nonmedical use of ketamine has steadily increased worldwide. "Special K", mainly made of ketamine, is one of the top three most abused synthetic drugs in China, causing severe individual and social problems[2,3].

The psychostimulant and reinforcing effects of ketamine are thought to be attributed to the inhibition of NMDA receptors on GABAergic neurons, as well as the disinhibition of excitatory neurons, causing excessive glutamate and dopamine release in the prefrontal cortex (PFC) and limbic striatal regions[4–6]. Mechanistic studies have suggested that several molecules and signaling cascades are involved in the neurochemical mechanisms of ketamine dependence, such as extracellular signal-regulated kinase (ERK), cAMP-responsive element binding protein (CREB), brain-derived neurotrophic factor, and glycogen synthase kinase 3β, as well as changes in membrane receptors[7,8]. Despite these advances, the downstream effects of ketamine have not been fully elucidated.

Addiction is characterized by heightened reinforcement from drug-paired stimuli[9]. The endocannabinoid system (ECS) plays a crucial role in regulating the drug reinforcement and the neurobehavioral effect of abuse substances, such as cocaine, morphine, heroin, and methamphetamine[10–14]. The ECS comprises G-protein-coupled receptors (CB1R and CB2R) and small neuromodulatory lipid ligands (N-arachidonoyl ethanolamide and 2-arachidonoylglycerol (2-AG)), as well as the biosynthetic and metabolic enzymes that synthesize and degrade these ligands, respectively[13]. Endocannabinoids (eCBs) mediate retrograde synaptic signaling to suppress neurotransmitter release at excitatory and inhibitory synapses, with both short- and long-term effects[15]. Activation of eCB signaling suppresses both excitatory and inhibitory signaling within some neuronal circuits[16], whereas blockade of eCB signaling attenuates addiction-related behaviors[17–20]. In addition, 2-AG and CB1R mediate neuronal morphology and growth, suggesting that eCB is involved in neuroplasticity[10,21]. Furthermore, chronic ketamine exposure modulates synaptic plasticity by increasing the dendritic spine density[22–24]. However, the contribution of eCBs to ketamine-seeking behaviors and neuroplasticity has remained largely unknown.

PR domain proteins (PRDMs) are a subfamily of kruppel-like zinc-finger gene products that are known to be involved in cell differentiation and growth[25]. PRDM5, an identified member of the PRDM family, is a sequence-specific transcriptional repressor encompassing an N-terminal PR domain without histone methyltransferase activity and C2H2 zinc-finger domains that mediate protein–DNA interactions[26]. A study showed that PRDM5 may function as a putative transcriptional repressor of the Mgll gene encoding monoacylglycerol lipase (MAGL)[27], and play a role in central nervous system pathophysiology after spinal cord injury[28].

In the present study, we show that eCBs in the dorsal striatum (caudate nucleus and putamen, CPu) play a critical role in ketamine addiction, and that blockade of CB1R signaling or overexpression of MAGL attenuates ketamine-induced behaviors and dendritic remodeling. PRDM5 functions as a transcriptional repressor of MAGL, which governs the hydrolysis of 2-AG in response to ketamine. Genetic manipulation of Mgll or Prdm5 prevents ketamine-induced behaviors by reducing the 2-AG level in the CPu.

## Results

**Ketamine significantly elevates the 2-AG level in the dorsal striatum.** To delineate changes in the lipidome of the brain after ketamine exposure, we initially performed untargeted lipidomic analysis of the brains of mice that received an intraperitoneal (i. p.) injection of 15 mg/kg ketamine or saline for 7 consecutive days. The dose of 15 mg/kg ketamine in mice is almost equivalent to recreational use in humans and is similar to the dose applied in other animal studies[29–31]. The PFC, nucleus accumbens (NAc), CPu (dorsal striatum), and hippocampus (Hipp), which are critically involved in various stages of the addiction cycle[32], were collected for untargeted lipidomic analysis using liquid chromatography coupled with tandem mass spectrometry (LC/MS–MS). Considering the important role of emotion in drug use vulnerability and the close link between the amygdala and emotion[33], we also collected the central nucleus of the amygdala (ACe) for lipidomic analysis. The major lipid classes were separated well with high resolution (Supplementary Fig. 1a). Well-fitted orthogonal projections to latent structures discriminant analysis models were constructed, and clear separations for each treatment group were obtained in most brain regions (Supplementary Fig. 1b, c). The apparent separation illustrated that the saline group and ketamine group had profoundly different lipid profiles. The CPu presented the most obvious lipid alterations, with changes in 174 lipids in 30 subclasses, and the Hipp showed the fewest lipid alterations (Fig. 1a), indicating that ketamine may preferentially affect the CPu. The detailed lipid alterations are shown in Supplementary Data 1 and Data 2, and the top ten lipids with the most significant elevation in each brain region are listed (Supplementary Fig. 1d). Although several lipids, such as phosphatidylcholines (PCs) and PEs, decreased evidently in the CPu in the ketamine group compared with the saline group (Supplementary Data 1), their alterations varied in different brain regions. Interestingly, a few lipid molecules of eCBs exhibited brain region-specific changes. Ketamine-treated mice showed significantly higher levels of anandamide (AEA) in the CPu, ACe, and PFC than saline-treated mice; moreover, 2-AG levels were obviously higher in the CPu and NAc of ketamine-treated mice (Fig. 1b). In light of the critical roles of eCBs in drug reward, we focused on their potential function in mediating ketamine-seeking behaviors.

Considering the identification uncertainty in untargeted lipidomic analysis, we performed targeted ultra-performance liquid chromatography tandem mass spectrometry (UPLC/MS/MS) to quantitatively validate the changes in eCBs in the CPu, Hipp, PFC, and NAc in two ketamine addiction models: locomotor activity and self-administration. We further used deuterated AEA (d4-AEA), deuterated 2-AG (d5-2-AG), and deuterated palmitoylethanolamide (PEA) as internal standards for comparison with these altered lipids and confirmed that the alterations in 2-AG and AEA were specific (Supplementary Fig. 2a–d). However, PEA, another ligand of eCB, was below the detection limit. We therefore focused on AEA and 2-AG in the following studies.

The locomotor activity paradigm is used to measure the psychomotor and psychostimulant effects of a drug by assessing moving distance after drug exposure. Compared to the saline group, the ketamine (15 mg/kg, i.p.) group exhibited a significant increase in moving distance (Supplementary Fig. 3a–c), demonstrating the psychostimulant and psychomotor properties of ketamine. At the end of the test, we quantitatively measured AEA and 2-AG levels in different brain regions by LC/MS–MS. Consistent with the results from the untargeted lipidomic analysis, the 2-AG levels were markedly increased by ~50% in the CPu and Hipp of ketamine-treated mice compared with those

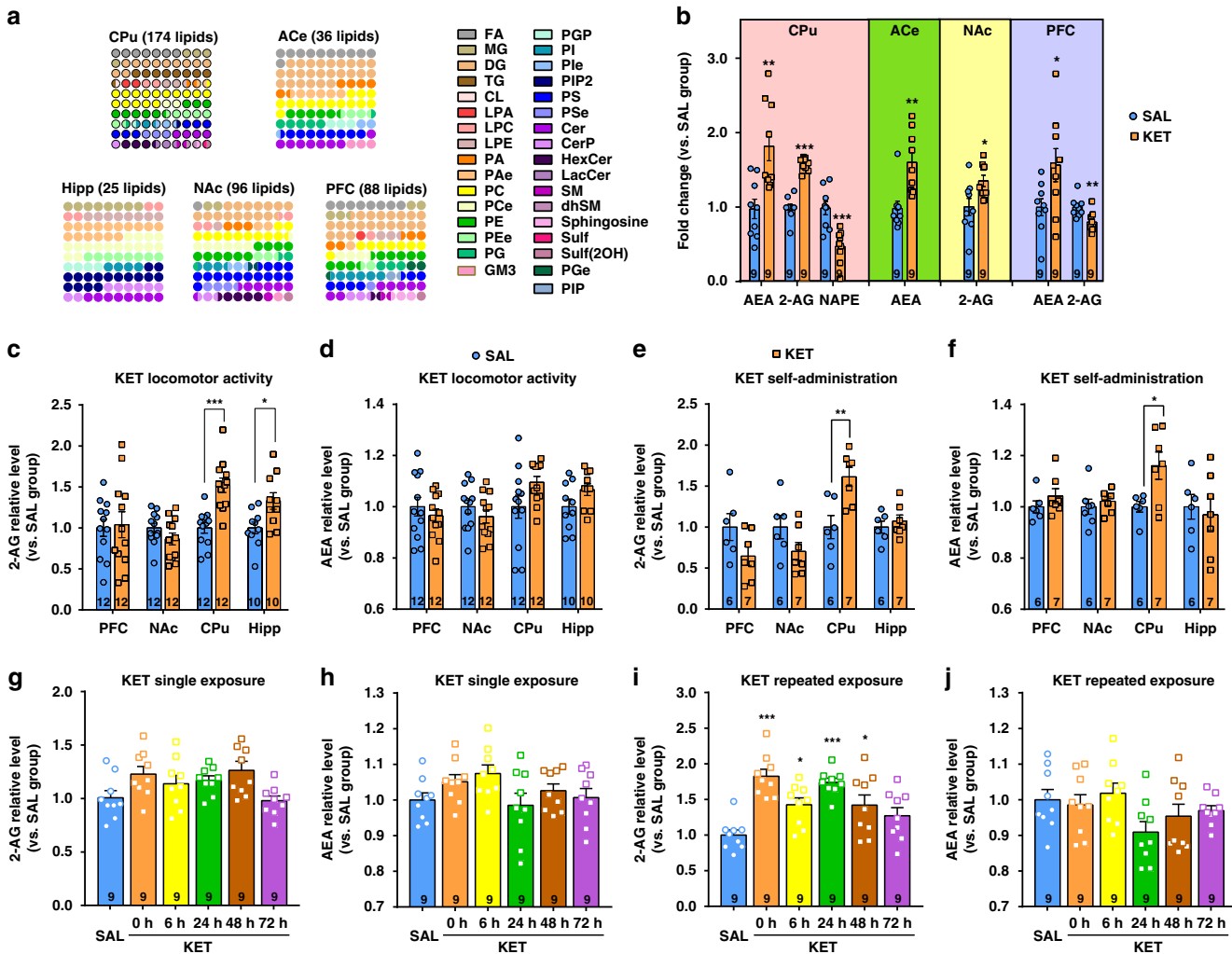

**Fig. 1 2-AG level in the dorsal striatum is significantly elevated by ketamine. a** Ketamine alters the lipid composition of brain in mice. Dot-plot graphic was adopted to exhibit the altered lipid subclass. The color of the dots represents a lipid subclass and the quantity of the dots represents the percentage. The lipids of CPu showed the most obvious alterations with 174 modified lipids. **b** Endocannabinoids changed significantly in the brain. The lipid level was normalized automatically by QI software and recalculated the relative levels compared to the saline control group (unpaired two-tailed t test, AEA in CPu $t(16) = 3.562$, $P = 0.0026$; 2-AG in CPu $t(16) = 9.877$, $P < 0.0001$; NAPE in CPu $t(16) = 4.413$, $P = 0.0004$; AEA in ACe $t(16) = 3.678$, $P = 0.002$; 2-AG in NAc $t(16) = 2.558$, $P = 0.0211$; AEA in PFC $t(16) = 2.264$, $P = 0.0378$; 2-AG in PFC $t(16) = 3.239$, **$P = 0.0051$). **c** Ketamine (15 mg/kg) significantly elevated 2-AG level both in the CPu and hippocampus of mice under hyperlocomotion paradigm (unpaired two-tailed t test, $t(22) = 4.805$, $P < 0.0001$; $t(18) = 2.666$, $P = 0.0158$). **d** Ketamine (15 mg/kg) did not change the AEA level in the brain of mice under hyperlocomotion. **e** Ketamine self-administration (0.5 mg/kg/infusion) significantly elevated 2-AG level in the CPu (unpaired two-tailed t test, $t(11) = 3.378$, $P = 0.0062$). **f** Ketamine self-administration (0.5 mg/kg/infusion) significantly increased AEA level in the CPu (unpaired two-tailed t test, $t(11) = 2.626$, $P = 0.0236$). **g, h** The levels of 2-AG and AEA were not altered after single ketamine exposure (15 mg/kg). **i** 2-AG level was significantly increased after repeated ketamine exposure (15 mg/kg; one-way ANOVA, followed by Dunnett's multiple comparisons test, $F(5,48) = 9.109$, $P < 0.0001$; 0 h vs. SAL, $P < 0.0001$; 6 h vs. SAL, *$P = 0.0187$; 24 h vs. SAL, $P < 0.0001$; 48 h vs. SAL, $P = 0.0207$). **j** AEA level was not altered after repeated ketamine exposure (15 mg/kg). Data are shown as mean ± SEM. Compared to SAL group, *$P < 0.05$, **$P < 0.01$, ***$P < 0.001$. SAL saline, KET ketamine, ACe central amygdaloid nucleus, CPu caudate nucleus and putamen, NAc nucleus accumbens, PFC prefrontal cortex, Hipp hippocampus, FA fatty acid, MG monoacylglycerol, DG diacylglycerol, TG triacylglycerol, CL cardiolipin, LPA lysophosphatidic acid, LPC lysophosphatidylcholine, LPE lysophosphatidylethanolamine, PA phosphatidic acid, PAe ether phosphatidic acid, PC phosphatidylcholine, PCe ether phosphatidylcholine, PE phosphatidylethanolamine, PEe ether phosphatidylethanolamine, PG phosphatidylglycerol, GM3 monosialodihexosylganglioside, PGP phosphatidylglycerol phosphate, PI phosphatidylinositol, PIe ether phosphatidylinositol, PIP2 phosphatidylinositol bisphosphate, PS phosphatidylserine, PSe ether phosphatidylserine, Cer ceramide, CerP ceramide 1-phosphate, HexCer hexosylceramide, LacCer lactosylceramide, SM sphingomyelin, dhSM dehydrosphingomyelin, Sulf sulfatides, Sulf(2OH) 2-hydroxy N-acyl sulfatide, PGe ether phosphatidylglycerol, PIP phosphatidylinositol phosphate. Source data provided as a Source Data file.

of saline-treated mice (Fig. 1c). However, ketamine did not significantly increase AEA level in the CPu despite an increasing tendency (Fig. 1d).

We next used a self-administration model to investigate whether eCBs are involved in ketamine-seeking behaviors. Mice self-administered intravenous ketamine (0.5 mg/kg/infusion)

using a fixed ratio 1 (FR1) schedule of reinforcement in operant chambers containing both active and inactive ports. Successful nose pokes of the active port triggered a ketamine infusion and activated a cue light, whereas nose pokes of the inactive port had no consequences. Compared to the saline group, the ketamine group exhibited a significant increase in active nose pokes that

was maintained for 3 consecutive days (Supplementary Fig. 3d, e), showing the reinforcing property of ketamine. Notably, the levels of both 2-AG and AEA were significantly higher in the CPu of ketamine self-administered mice than in that of saline-treated mice (Fig. 1e, f). In these two behavioral models, both 2-AG and AEA levels increased significantly, and the augmentation of 2-AG was higher than that of AEA, suggesting the potential effect of these two lipids on the behavioral effects of ketamine.

We further investigated the acute and chronic effects of ketamine on 2-AG and AEA production in the CPu. The levels of 2-AG and AEA were measured 0, 6, 24, 48, and 72 h after a single ketamine injection (15 mg/kg, i.p.) or after the last injection in the repeated ketamine treatment (15 mg/kg, i.p., consecutive injection for 7 days). The levels of 2-AG were significantly increased 0, 6, 24, and 48 h after the last ketamine injection compared with the saline-injected group (Fig. 1i). Intriguingly, the 2-AG level returned to the baseline level at 72 h (Fig. 1i), indicating that the production of 2-AG was reversible. However, a single ketamine exposure did not alter the 2-AG level (Fig. 1g). Unexpectedly, AEA levels were not significantly altered by either single or repeated ketamine administration (Fig. 1h, j), suggesting that AEA may not be involved in the psychomotor effect of ketamine.

Taken together, these results show that different ketamine paradigms are capable of remodeling the lipidome in the CPu and that 2-AG appears to be the lipid molecule predominantly modulated by ketamine, suggesting a role for 2-AG in mediating the ketamine effect. Moreover, the CPu may be the key brain region affected by ketamine.

**CB1R blockade attenuates the psychostimulant and reinforcing effects of ketamine**. CB1Rs are the dominant receptors of 2-AG and the most abundant G-protein-coupled receptors in the CPu[13]. Considering that ketamine significantly elevated the 2-AG level in the above lipidomic studies, we speculated that 2-AG-CB1R signaling may mediate ketamine-induced behaviors. To this end, we used rimonabant, a selective CB1R inverse agonist, to explore the role of CB1Rs in mediating the psychostimulant and reinforcing effects of ketamine.

First, we tested the effect of rimonabant, through systemic or intracranial delivery, on ketamine-induced hyperlocomotor activity. The mice were administered rimonabant i.p. half an hour before ketamine injection. Rimonabant (0.6 mg/kg) did not affect the baseline locomotor activity, but effectively blocked ketamine-induced hyperactivity (Supplementary Fig. 4a–c). Because CB1Rs are highly expressed in the dorsal lateral striatum (DLS) rather than the dorsal medial striatum[34], we chose the DLS as the target region for intracranial delivery of rimonabant. Rimonabant (0.6 μg per side) was bilaterally infused into the DLS through preimplanted cannulas (Supplementary Fig. 5). After a week of recovery from cannulation surgery, the locomotor activity of mice was measured daily for 7 consecutive days (Fig. 2a), and rimonabant was delivered into the DLS half an hour before each ketamine injection. Consistent with the results from systemic delivery, intra-DLS infusion of rimonabant clearly reduced ketamine-induced hyperactivity (Fig. 2b), suggesting that DLS is the critical brain region involved in ketamine-induced hyperactivity.

To elucidate the downstream intracellular pathways involved in the effects of CB1R blockade, we used immunoblotting to study the expression of protein kinase A (PKA), ERK, phosphorylated ERK (pERK), CREB, and pCREB, key regulators of synaptic plasticity. Considering that cAMP signaling is negatively regulated by CB1R activation, we quantitatively measured cAMP level and the expression of cAMP signaling proteins in the DLS.

Importantly, the results showed that ketamine clearly reduced the cAMP level in the DLS in the hyperlocomotion model, suggesting CB1R activation; however, this effect was significantly attenuated by intra-DLS injection of rimonanbant (Fig. 2c and Supplementary Fig. 6). Interestingly, ketamine increased pERK expression, but decreased the expression of CREB and pCREB (Fig. 2d, e). Such effects were also markedly reversed by rimonabant (Fig. 2d, e), indicating that blockade of CB1R signaling is capable of attenuating ketamine-induced hyperactivity and synaptic plasticity.

Next, the effect of CB1R blockade through the systemic or intracranial delivery of rimonabant on ketamine reinforcement was investigated in an intravenous self-administration paradigm. Both systemic administration and bilateral intra-DLS infusion of rimonabant effectively reduced active nose pokes and drug injections (Supplementary Fig. 7a, b and Fig. 2f–h), indicating an inhibitory effect of rimonabant on the ketamine reinforcing effect. Interestingly, ketamine decreased PKA expression, but increased both CREB and pCREB expression in the CPu, indicating that kinases other than PKA may contribute to CREB phosphorylation (Fig. 2i, j). In addition, rimonabant effectively counteracted the effects of ketamine (Fig. 2i, j), indicating an inhibitory effect of rimonabant on volitional ketamine-taking behavior. As the effects of ketamine on PKA and CREB phosphorylation differed in the hyperlocomotion and self-administration paradigms, we speculated that it may be due to the combination of psychopharmacological effects and learning induced by ketamine.

**Ketamine remodels dendrite structure through CB1Rs in the DLS**. Drug-induced structural changes in neurons play a critical role in mediating the long-lasting behavioral adaptations caused by the repetitive administration of drugs of abuse[35]. We thus investigated the effect of CB1R blockade on dendritic remodeling in the DLS in mice with ketamine hyperlocomotion (15 mg/kg). Golgi–Cox staining was conducted to observe the morphological changes in the spiny projection neurons (SPNs) in the DLS. The DLS was well impregnated, and the glial cells were recognized by the typical morphological profile of tufted clumps (Supplementary Fig. 8a, shown in the blue circle). The vast majority of SPNs had medium-sized soma and spine-laden dendrites, with a gradual increase in spine density toward the more distal segments. These cells were considered SPNs and used for quantitative analysis (Supplementary Fig. 8a, shown in the red square). We analyzed the dendritic complexity of SPNs using ImageJ and the Sholl analysis plugin, as described previously[36,37]. Dendrites were traced, the lengths of the dendrites were measured by ImageJ software (Fig. 3a), and concentric ring intersections were identified (Supplementary Fig. 8b). Ketamine significantly enhanced dendritic complexity, including promoting dendritic length and spine density (Fig. 3b–e). Rimonabant partly inhibited ketamine-induced dendritic arborization (Fig. 3b), but did not reduce the total dendritic length promoted by ketamine (Fig. 3c). In addition, rimonabant obviously reversed the ketamine-induced increase in spine density (Fig. 3d, e). We finally performed immunoblotting to analyze the expression of synaptic-related proteins. Ketamine obviously enhanced the expression of phosphorylated cofilin (pCofilin), GluA1 (phospho-Ser 831) (pSer831-GluA1), GluA1 (phospho-Ser 845) (pSer845-GluA1), and GluN1; nevertheless, these effects were clearly attenuated by an intra-DLS infusion of rimonabant (Fig. 3f, g). Collectively, our results indicate that CB1Rs contribute to ketamine-induced dendritic remodeling and synaptic plasticity in the DLS.

**Ketamine elevates the 2-AG level by repressing MAGL expression**. To determine how ketamine increases 2-AG

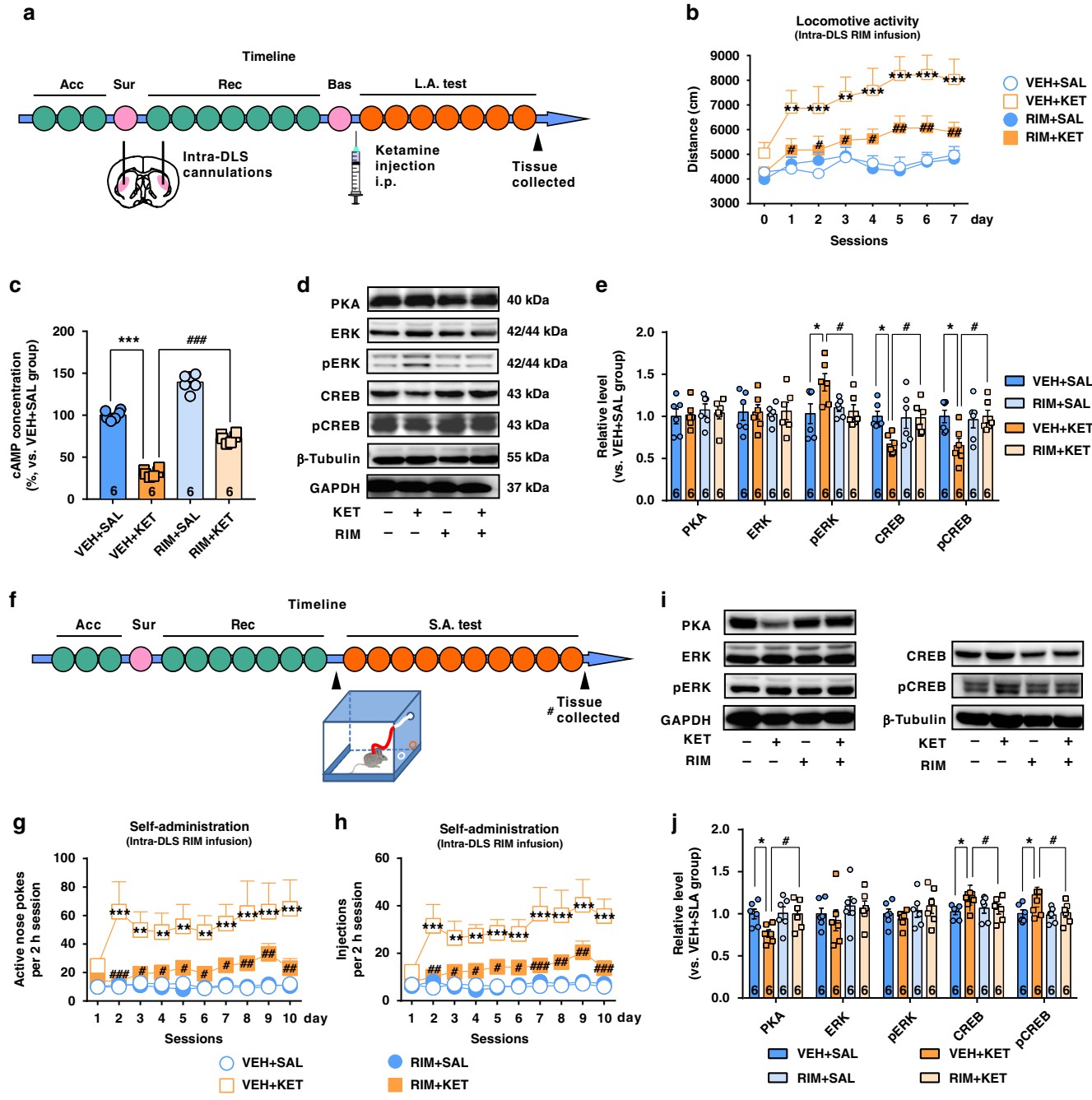

production in vivo, we first applied qRT-PCR to measure the mRNA expression of all genes involved in 2-AG metabolism in various brain regions with the ketamine hyperlocomotion paradigm, including 2-AG synthetases (diacylglycerol lipase (DAGL)$_\alpha$ and DAGL$_\beta$) and catabolic enzymes (MAGL, ABDH6, and ABDH12). Notably, the mRNA expression of MAGL, which hydrolyzes >85% of 2-AG[38], was significantly decreased by ketamine in the CPu and PFC, but not in the ventral tegmental area or Hipp (Fig. 4a). However, the mRNA expression levels of other 2-AG-metabolizing enzyme genes, including DAGL$_\alpha$, DAGL$_\beta$, ABDH6, and ABDH12, were not obviously altered by ketamine (Fig. 4a). Using immunoblotting, we confirmed that ketamine treatment significantly decreased the protein expression of MAGL in the CPu (Fig. 4b, c), but not in the PFC (Supplementary Fig. 9a), suggesting that ketamine elevated the 2-AG

level by reducing 2-AG hydrolysis. We also measured the protein expression of CNR1 (CB1R) and AEA-metabolizing enzymes (NAPE-PLD and FAAH) by immunoblotting analysis. As expected, no obvious changes were observed in these proteins (Supplementary Fig. 9b). These results indicate that ketamine specifically downregulates MAGL expression in the CPu.

We further assayed the direct effect of ketamine on MAGL activity using an in vitro detection method. JZL195, a known antagonist of MAGL, was used as a positive control[39]. JZL195 dose dependently decreased MAGL activity (Supplementary Fig. 10), whereas ketamine failed to do so (Fig. 4f), indicating that ketamine upregulates 2-AG level through a transcriptional regulatory mechanism of MAGL, but not through a direct inhibitory effect on MAGL activity. Thus, the ketamine-induced increase in 2-AG may be predominantly attributed to the

**Fig. 2 CB1Rs blockage attenuates the psychostimulant and reinforcing effects of ketamine. a** Experimental time course for locomotor activity detection in the mice receiving intra-DLS infusion of rimonabant (RIM). **b** Bilateral intra-DLS infusion of RIM (0.6 μg per side) attenuated ketamine-induced hyperactivity ($n = 10$ mice/group, two-way repeated measured ANOVA, followed by Dunnett's multiple comparisons test, Drug, $F_{(3,36)} = 10.22$, $P < 0.0001$; time, $F_{(7,252)} = 10.12$, $P < 0.0001$; interaction, $F_{(21,252)} = 1.953$, $P = 0.0088$). **c** Intra-DLS infusion of RIM significantly elevated ketamine-reduced cAMP in hyperlocomotion model. (One-way ANOVA, followed by Dunnett's multiple comparisons test, $F_{(3,20)} = 283.6$, $P < 0.0001$). **d** Representative immunoblotting of downstream signaling molecules after CB1Rs blockage in ketamine-induced hyperlocomotion. **e** Statistical analysis of immunoblotting showed that in hyperlocomotion model, ketamine increased pERK expression, but decreased CREB and pCREB expression; these effects were reversed by rimonabant (one-way ANOVA, followed by Dunnett's multiple comparisons test, pERK, $F_{(3,20)} = 3.639$, $P = 0.0304$; CREB, $F_{(3,20)} = 3.802$, $P = 0.0263$; pCREB, $F_{(3,20)} = 3.933$, $P = 0.0234$). The optical densities of the detected proteins are normalized to the mean of GAPDH and α-tubulin. **f** Experimental time course for self-administration in mice receiving intra-DLS RIM infusion. **g** Bilateral intra-DLS infusion of RIM (1 μM, 1 μL per side) effectively reduced ketamine-enhanced active nose pokes ($n = 7$ mice/group, two-way repeated measured ANOVA, followed by Dunnett's multiple comparisons test, treatment, $F_{(3,24)} = 7.477$, $P = 0.0011$; time, $F_{(9,216)} = 3.383$, $P = 0.0007$; interaction, $F_{(27,216)} = 1.901$, $P = 0.0065$). **h** Bilateral intra-DLS infusion of RIM effectively decreased drug injections ($n = 7$ mice/group, two-way repeated measured ANOVA, followed by Dunnett's multiple comparisons test, treatment, $F_{(3,24)} = 10.86$, $P = 0.0001$; time, $F_{(9,216)} = 5.055$, $P < 0.0001$; interaction, $F_{(27,216)} = 2.172$, $P = 0.0012$). **i** Representative immunoblotting of downstream signaling molecules after CB1Rs blockage in ketamine self-administration test. **j** Statistical analysis for immunoblotting showed that ketamine decreased PKA expression, but increased CREB and pCREB expression; rimonabant counteracted these effects (one-way ANOVA, followed by Dunnett's multiple comparisons test: PKA, $F_{(3,20)} = 3.37$, $P = 0.0389$; CREB, $F_{(3,20)} = 3.546$, $P = 0.0331$; pCREB, $F_{(3,20)} = 3.656$, $P = 0.03$). Data are shown as mean ± SEM. Compared to VEH + SAL group, *$P < 0.05$, **$P < 0.01$, ***$P < 0.001$; Compared to VEH + KET group, #$P < 0.05$, ##$P < 0.01$. SAL saline, KET ketamine, VEH vehicle, RIM rimonabant, Acc acclimation, Sur surgery, Rec recovery, Bas baseline test, L.A. test locomotor activity test, S.A. test self-administration test. One small circle represents for a day. One big circle represents for a week. Source data provided as a Source Data file.

decreased expression of MAGL, the key hydrolytic enzyme for 2-AG. Through immunochemistry, we identified the cell type in which MAGL expression was altered by ketamine. Antibodies against NeuN and glial fibrillary acidic protein (GFAP) were used to label neurons and astrocytes, respectively. Ketamine clearly decreased MAGL expression in DLS neurons, as evidenced by NeuN staining (Fig. 4d, e). As >95% of neurons in the DLS are SPNs[40], we assumed that the ketamine-induced downregulation of MAGL may mainly occur in the SPNs.

To explore the time-dependent effect of ketamine on the expression of 2-AG-metabolizing enzymes, we measured MAGL and DAGL$_α$ expression in the DLS at different time points after ketamine exposure. There was no change in MAGL and DAGL$_α$ expression in response to a single ketamine challenge (Fig. 4g, h and Supplementary Fig. 11a). Nevertheless, MAGL expression was significantly downregulated after 7 days of repeated ketamine injection, and this downregulation was maintained for 72 h (Fig. 4i, j). Interestingly, the expression of the 2-AG synthetase DAGL$_α$ was not affected within 48 h, but was reduced at 72 h after 7 days of repeated ketamine injection (Supplementary Fig. 11b). As the 2-AG level returned to the baseline level at 72 h (Fig. 1i), the downregulated DAGL$_α$ expression level may reflect a compensatory mechanism for 2-AG metabolism regulation.

Finally, we explored whether NMDAR is involved in the effect of ketamine on the ECS. Mice received an i.p. injection of 0.3 mg/kg MK801, an NMDAR antagonist, for 7 consecutive days. We found that MK801 clearly induced hyperlocomotion in mice (Supplementary Fig. 12a). Furthermore, MK801 also significantly elevated 2-AG and AEA levels, and decreased MAGL expression in the CPu (Supplementary Fig. 12b–e). These effects of MK801 were similar to those of ketamine, suggesting that NMDAR is involved in the effect of both ketamine and MK801 on the ECS.

**MAGL is essential for ketamine-induced hyperlocomotor and ketamine-seeking behaviors**. To explore how MAGL impacts the addiction-related behaviors associated with ketamine, we investigated the effect of MAGL overexpression in the DLS on ketamine-induced hyperlocomotor activity and ketamine-seeking behaviors. We generated a long-lasting and neurotropic viral vector (adeno-associated virus 2/8, abbreviated AAV2/8) that expressed MAGL (pAAV-CAG-EGFP-2A-Mgll-3FLAG,

abbreviated AAV-*Mgll*) or eGFP (pAAV-CAG-EGFP-2A-MCS-3FLAG, abbreviated AAV-eGFP) alone. The design details for the virus are shown in Supplementary Fig. 13. We bilaterally injected AAV-*Mgll* or AAV-eGFP into the DLS, and measured MAGL mRNA and protein expression 3 weeks later. FLAG-eGFP was well expressed in the DLS, indicating the accurate injection of the viral vector and MAGL expression (Fig. 5a). Immunoblotting and quantitative PCR (qPCR) analysis further demonstrated the successful expression of MAGL in the DLS (Fig. 5b, c).

We first tested the effect of MAGL overexpression on the locomotor activity of mice. Compared with the AAV-eGFP control, AAV-*Mgll* significantly reduced the moving distance stimulated by 15 mg/kg ketamine (Fig. 5e, f). We then conducted a rescue test to investigate the effect of 2-AG supplementation on AAV-*Mgll*-attenuated locomotor activity. Different dosages of 2-AG (0.4, 2, and 10 μM) were bilaterally infused into the DLS (1 μL per side) through preimplanted catheters 30 min before locomotor activity detection. Importantly, 2-AG obviously restored the psychostimulant effect of ketamine in the mice treated with AAV-*Mgll* (Fig. 5g, h) but showed no effect on baseline locomotion activity in normal mice (Supplementary Fig. 14). These results indicated that MAGL may modulate ketamine-induced hyperlocomotor activity by regulating 2-AG hydrolysis.

We further investigated whether MAGL overexpression attenuates the reinforcing effect of ketamine in the self-administration paradigm. Two weeks after the intra-DLS infusion of AAV-*Mgll* or AAV-eGFP, a catheter was inserted into the right jugular vein of mice, and then the mice continued to recover for 1 week before the self-administration test (Fig. 5i). AAV-*Mgll* clearly blocked the acquisition of ketamine self-administration with a significant decrease in active nose pokes and drug injection (Fig. 5j, k). Intriguingly, intra-DLS injection of 2-AG did not rescue the inhibitory effect of MAGL overexpression on ketamine reinforcing behavior (Fig. 5j, k), suggesting that there are other factors involved in volitional drug-taking initiation. We then continued to investigate whether 2-AG modulates the sensitivity of the reinforcing effect of ketamine. Mice were trained to self-administer ketamine for at least 15 days as previously described[41], and the effect of 2-AG on ketamine self-administration was measured in sequential self-administration tests on an FR

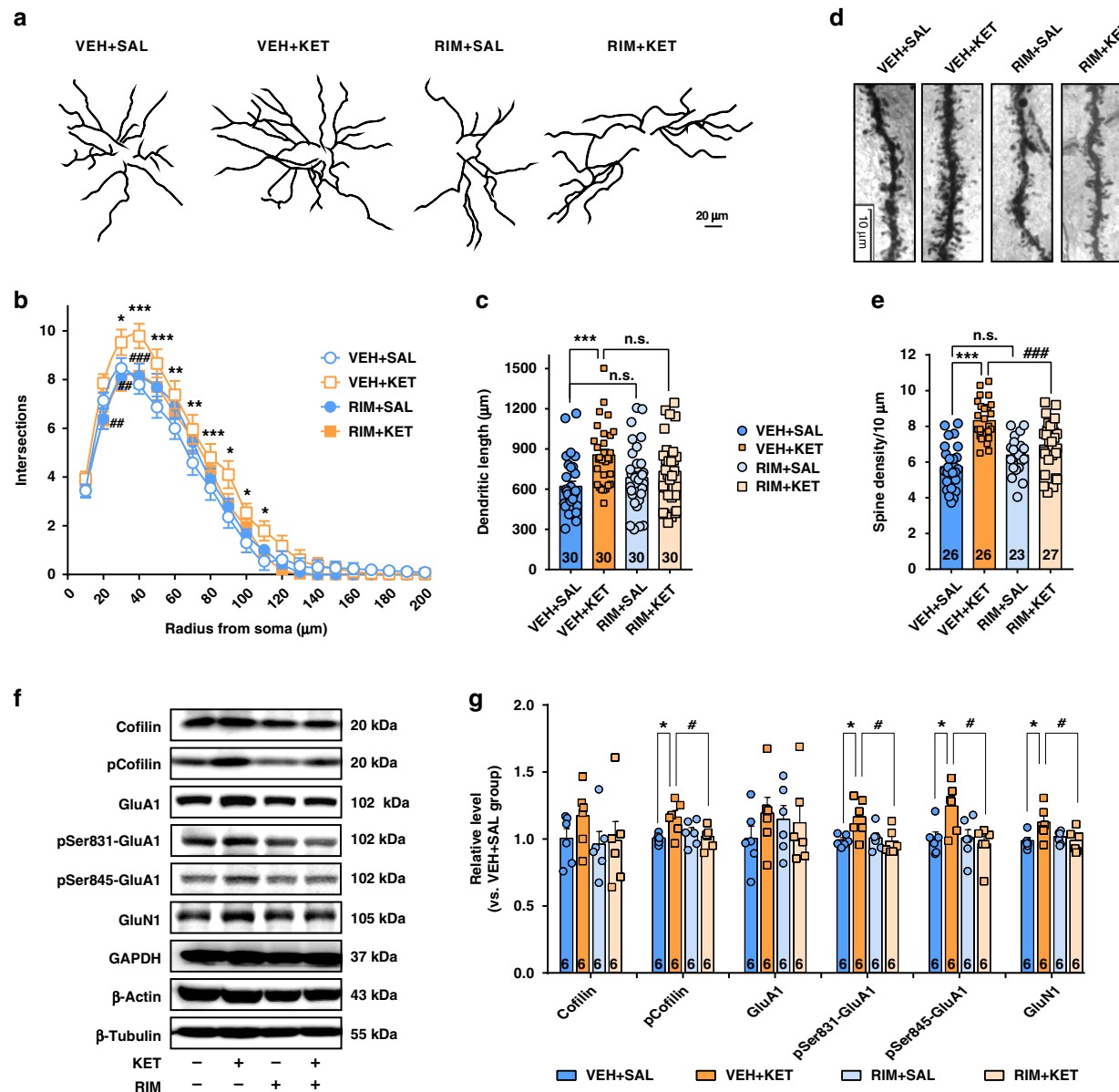

**Fig. 3 Ketamine remodels dendrite structure in the DLS in hyperlocomotion paradigm by a CB1R-dependent manner. a** Representative traces of Golgi–Cox-stained DLS SPNs. **b** CB1R blockage partly inhibited ketamine-induced dendritic arborization ($n = 30$ neurons/group, two-way repeated measured ANOVA, followed by Tukey's multiple comparisons test, treatment, $F(3,87) = 3.203$, $P = 0.0272$; radius, $F(19,551) = 427.7$, $P < 0.0001$; interaction, $F(57,1653) = 1.554$, $P = 0.0056$). **c** CB1R blockage did not reduce dendritic length promoted by ketamine (one-way ANOVA, followed by Tukey's multiple comparisons test, $F(3,116) = 5.448$, $P = 0.0015$; VEH + SAL vs. VEH + KET, $P = 0.0009$). **d** Representative images for dendritic spines of DLS SPNs stained by Golgi–Cox (100× oil len). **e** CB1R blockage obviously attenuated spine density promoted by ketamine (one-way ANOVA, followed by Tukey's multiple comparisons test, $F(3,98) = 21.49$, $P < 0.0001$; VEH + SAL vs. VEH + KET, $P < 0.0001$; VEH + KET vs. RIM + KET, $P < 0.0001$;). **f** Representative immunoblotting for dendritic proteins. **g** Ketamine enhanced the expression of pCofilin, pSer831-GluA1, pSer845-GluA1, and GluN1; nevertheless, CB1R antagonist significantly attenuated these effects (one-way ANOVA, followed by Tukey's multiple comparisons test, pCofilin $F(3,20) = 4.095$, $P = 0.0203$; pSer831-GluA1, $F(3,20) = 4.705$, $P = 0.0121$; pSer845-GluA1, $F(3,20) = 4.479$, $P = 0.0146$; GluN1, $F(3,20) = 3.688$, $P = 0.0291$). Data are shown as mean ± SEM. Compared to VEH + SAL group, *$P < 0.05$, **$P < 0.01$, ***$P < 0.001$; compared to VEH + KET group, #$P < 0.05$, ##$P < 0.01$, ###$P < 0.001$. SAL saline, KET ketamine, 15 mg/kg; VEH vehicle, RIM rimonabant. Source data provided as a Source Data file.

schedule of ketamine reinforcement, which was gradually increased from 1 (FR1) to 5 (FR5) active nose pokes for a single infusion of ketamine. This procedure was followed by dose–response reinforcement schedules (Fig. 5m). Stabilized ketamine self-administration was counterbalanced across both study groups (Fig. 5n), and no group differences were apparent in dose–response curves during within-session testing in baseline

self-administration (Fig. 5o). Notably, mice treated with an intra-DLS infusion of 10 μM 2-AG exhibited a clear leftward shift in dose sensitivity for maintaining ketamine self-administration (Fig. 5p), indicating that 2-AG supplementation increased the sensitivity to ketamine reinforcement. Collectively, these results show that MAGL-2-AG signaling is necessary for the addiction-associated neurobehavioral effects of ketamine.

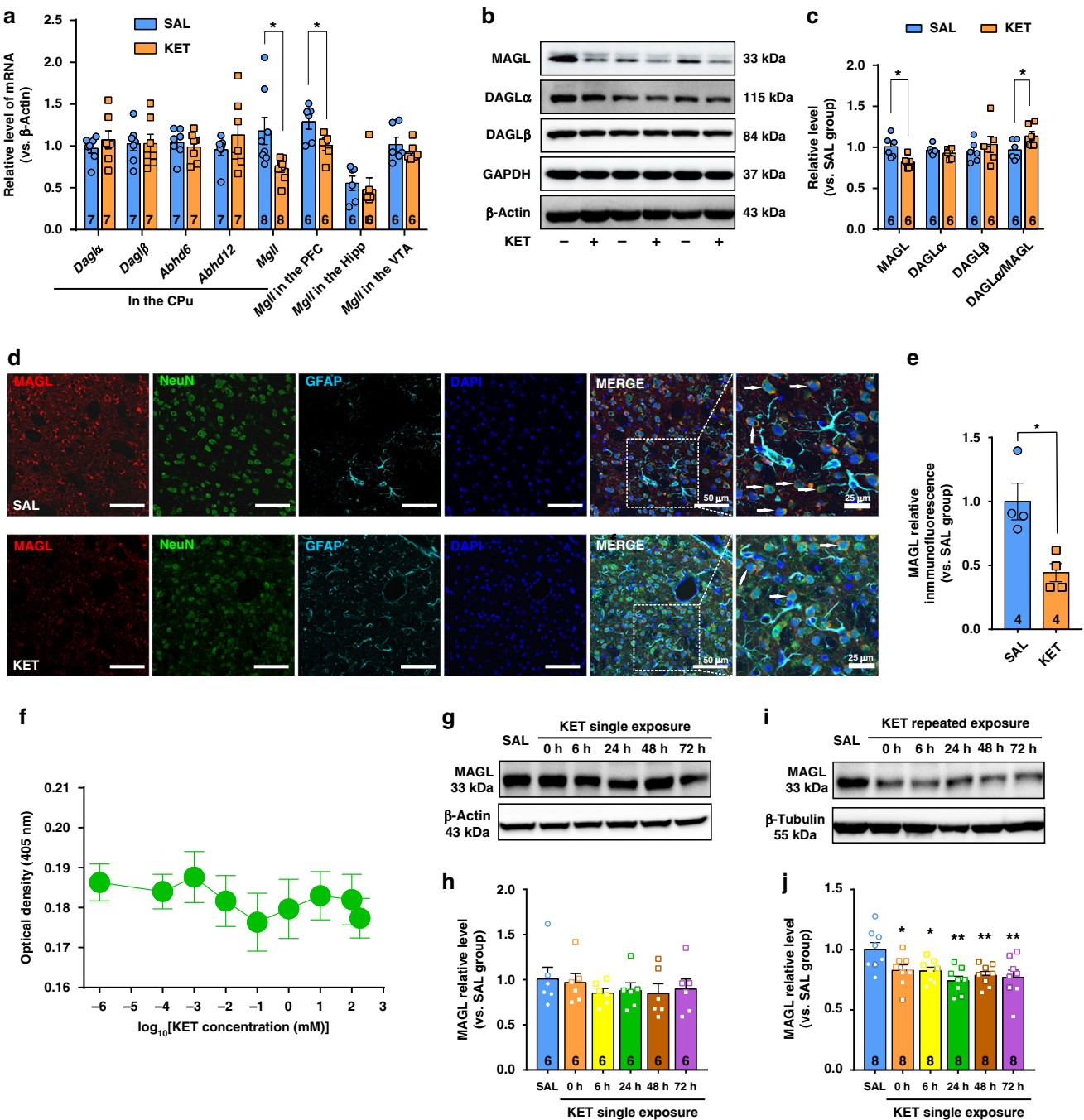

**Fig. 4 Ketamine represses MAGL transcription in the CPu, leading to an increase in 2-AG production. a** Ketamine (15 mg/kg) decreased mRNA expression of MAGL gene in the CPu and PFC (Unpaired $t$ test, $Mgll$ in the CPu, $t(14) = 2.674$, $P = 0.0182$; $Mgll$ in the PFC, $t(10) = 2.647$, $P = 0.0244$). **b** Representative immunoblotting for the expression of 2-AG metabolic enzymes in the dorsal striatum. **c** Ketamine (15 mg/kg) decreased MAGL protein expression and increased the ratio of DAGLα to MAGL (DAGLα/MAGL) in the CPu (Unpaired two-tailed $t$ test: MAGL, $t(10) = 3.069$, $P = 0.0119$; DAGLα/ MAGL, $t(10) = -2.324$, $P < 0.05$). **d** Co-immunostaining for MAGL (red), NeuN (green), GFAP (wathet blue), and DAPI (deep blue) in the DLS. Arrows indicate the neurons positive for MAGL. **e** Ketamine (15 mg/kg) reduced MAGL expression in the DLS neurons (unpaired two-tailed $t$ test, $t(6) = 3.486$, $P = 0.0131$). **f** Ketamine did not affect MAGL enzyme activity in vitro. $n = 3$ samples/group. **g, h** MAGL expression was not altered by a single ketamine (15 mg/kg). **i, j** MAGL expression decreased within 72 h after repeated exposure to ketamine (15 mg/kg; one-way ANOVA, followed by Dunnett's multiple comparisons test, $F(5,42) = 3.922$, $P = 0.0052$; 0 h vs. SAL, $P = 0.049$; 6 h vs. SAL, $P = 0.0404$; 24 h vs. SAL, $P = 0.0014$; 48 h vs. SAL, $P = 0.0099$; 72 h vs. SAL, $P = 0.0048$). Data are shown as mean ± SEM. Compared to SAL group, *$P < 0.05$, **$P < 0.01$. CPu caudate nucleus and putamen, PFC prefrontal cortex, Hipp hippocampus, VTA ventral tegmental area, SAL saline, KET ketamine. Source data provided as a Source Data file.

To confirm the effect of MAGL overexpression on the production of 2-AG and AEA, we quantitatively measured the levels of these two lipids using LC/MS–MS. Consistently, AAV-$Mgll$ effectively reduced 2-AG production in the DLS, and intra-DLS 2-AG infusion obviously increased 2-AG production both in ketamine hyperloco-motion and self-administration paradigms (Fig. 5d, l), indicating the successful manipulation of 2-AG production through the genetic regulation of MAGL expression. In addition, no apparent changes in

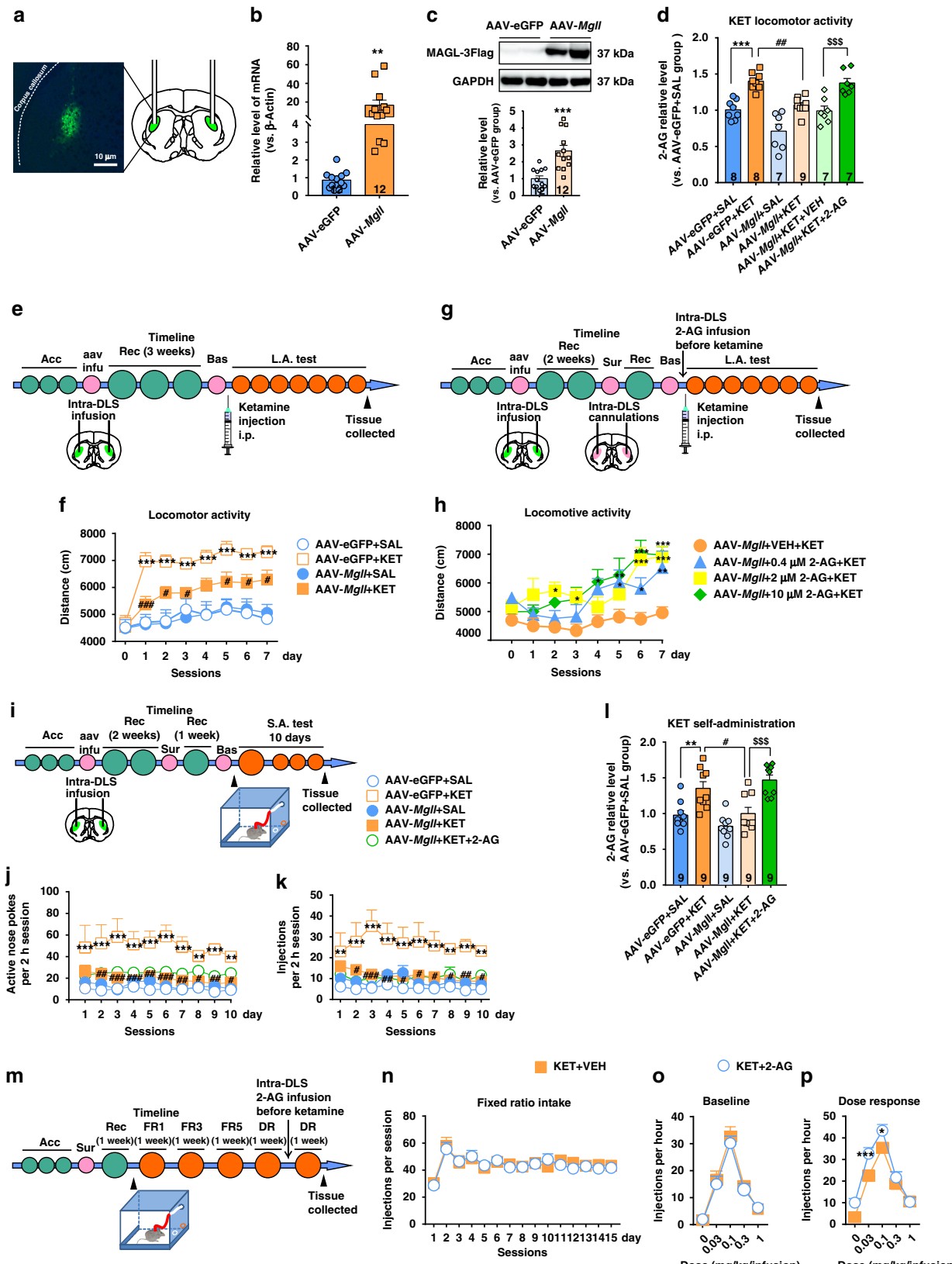

AEA levels were observed, indicating that MAGL overexpression specifically influences the level of 2-AG (Supplementary Fig. 15). Taken together, these results demonstrate that MAGL, acting as a key 2-AG hydrolysis enzyme, plays a central role in regulating the 2-AG level in the DLS, and is required for the psychostimulant and reinforcing effects of ketamine.

**PRDM5 regulates *Mgll* gene expression through a transcriptional mechanism**. By chromatin immunoprecipitation (ChIP)-seq screening, a previous study showed that MAGL might be a putative target gene of PRDM5 in ApcMin-driven intestinal adenomas[27]. We thus investigated the role of PRDM5 in the transcriptional regulation of the *Mgll* gene. We separated the

**Fig. 5 MAGL is essential for the psychostimulant and reinforcing effects of ketamine. a** Representative images of AAV targeting to mouse DLS. **b** AAV-*Mgll* significantly increased the transcription of *Mgll* gene (unpaired two-tailed *t* test, $t(22) = 2.889$, $P = 0.0085$). Compared to AAV-eGFP group, **$P <$ 0.01. **c** AAV-*Mgll* significantly increased the MAGL protein expression (unpaired *t* test, $t(22) = 4.411$, $P = 0.0002$). Compared to AAV-eGFP group, ***$P <$ 0.001. **d** AAV-*Mgll* effectively reduced the 2-AG production and intra-DLS infusion of 2-AG obviously increased 2-AG level in AAV-*Mgll*-infected mice under hyperlocomotion paradigm (one-way ANOVA, followed by Tukey's multiple comparisons test, $F(5,40) = 18.77$, $P < 0.0001$; AAV-eGFP + KET vs. AAV-eGFP + SAL, $P = 0.0003$; AAV-*Mgll* + KET vs. AAV-eGFP + KET, $P = 0.0019$; AAV-*Mgll* + KET + VEH vs. AAV-eGFP + KET + 2-AG, $P = 0.0009$). Compared to AAV-eGFP + SAL group, ***$P < 0.001$; compared to AAV-eGFP + KET group, ##$P < 0.01$; compared to AAV-*Mgll* + KET + VEH group, $$$$P < 0.001$. **e** Experimental time course for intra-DLS injection of AAV in ketamine hyperlocomotion test. **f** Intra-DLS injection of AAV-*Mgll* attenuated ketamine-evoked hyperlocomotion ($n = 11$ mice/group, two-way repeated measured ANOVA, followed by Dunnett's multiple comparisons test, treatment, $F(3,40) = 12.87$, $P < 0.0001$; time, $F(7,280) = 19.87$, $P < 0.0001$; interaction, $F(21,280) = 2.983$, $P < 0.0001$). Compared to AAV-eGFP + SAL group, ***$P < 0.001$; compared to AAV-eGFP + KET group, #$P < 0.05$, ###$P < 0.001$. **g** Experimental time course for intra-DLS infusion of 2-AG in AAV-*Mgll*-infected mice under hyperlocomotion condition. **h** Intra-DLS infusion of 2-AG obviously restored the psychostimulant effect of ketamine in mice infected with AAV-*Mgll* ($n = 11$ mice/group, two-way repeated measured ANOVA, followed by Dunnett's multiple comparisons test, treatment $F(3,40) = 5.787$, $P = 0.0022$; time $F(7,280) = 14.77$, $P < 0.0001$; interaction, $F(21,280) = 2.189$, $P = 0.0023$). Compared to AAV-*Mgll* + VEH + KET group, *$P < 0.05$, **$P < 0.01$, ***$P < 0.001$. **i** Experimental time course for intra-DLS injection of AAV and 2-AG in ketamine self-administration model. **j, k** Intra-DLS injection of AAV-*Mgll* reduced ketamine-seeking behaviors, but intra-DLS supplementation of 2-AG could not restore ketamine-induced behaviors in the AAV-*Mgll*-infected mice ($n = 10$ mice/group, two-way repeated measured ANOVA, followed by Tukey's multiple comparisons test; active nose pokes, virus $F(4,45) = 21.19$, $P < 0.0001$; time $F(9,405) = 0.8489$, $P = 0.5714$; interaction, $F(36,405) = 0.423$, $P = 0.9988$; injections, Virus $F(4,45) = 14.58$, $P < 0.0001$; time $F(9,405) = 0.6625$, $P = 0.7429$; interaction, $F(36,405) = 0.5889$, $P = 0.9731$). Compared to AAV-eGFP + SAL group, **$P < 0.01$, ***$P < 0.001$; compared to AAV-eGFP + KET group, #$P < 0.05$, ##$P < 0.01$, ###$P < 0.001$. **l** Overexpression MAGL in the DLS decreased 2-AG level and intra-DLS injection of 2-AG restored 2-AG level in the AAV-*Mgll*-infected mice under ketamine self-administration paradigm (one-way ANOVA, followed by Tukey's multiple comparisons test, $F(4,40) = 13.27$, $P < 0.0001$; AAV-eGFP + KET vs. AAV-eGFP + SAL, $P = 0.0088$; AAV-*Mgll* + KET vs. AAV-eGFP + KET, $P = 0.0156$; AAV-*Mgll* + KET vs. AAV-*Mgll* + KET + 2-AG, $P = 0.0007$). Compared to AAV-eGFP + SAL group, **$P < 0.01$; compared to AAV-eGFP + KET group, #$P < 0.05$; compared to AAV-*Mgll* + KET + VEH group, $$$$P < 0.001$. **m** Experimental time course for ketamine dose–response effect before and after intra-DLS infusion of 2-AG. **n, o** The baselines of self-administration and dose–response curves were counterbalanced between vehicle-infusion and 2-AG-infusion groups. $n = 6$ mice/group. **p** 2-AG increased the sensitivity to ketamine in self-administering mice ($n = 6$ mice/group, two-way repeated measured ANOVA, followed by Bonferroni's multiple comparisons test, treatment, $F(1,10) = 9.772$, $P = 0.0108$; ketamine dose $F(4,40) = 86.76$, $P < 0.0001$; interaction, $F(4,40) = 4.235$, $P = 0.006$). Compared to KET + VEH group, *$P < 0.05$, ***$P < 0.001$. Data are shown as mean ± SEM. SAL saline, KET ketamine, VEH vehicle, Acc acclimation, aav infu AAV infusion, Rec recovery, Bas baseline test, FR1 fixed ratio1, FR3 fixed ratio3, FR5 fixed ratio5, DR dose response, L.A. test locomotor activity test, S.A. test self-administration test. One small circle represents for a day and one big circle for a week. Source data provided as a Source Data file.

nuclear fraction and measured the nuclear expression of PRDM5 in the CPu of mice subjected to the hyperlocomotion paradigm (i.e. i.p. injection of 15 mg/kg ketamine for 7 consecutive days). Immunoblotting analysis showed that ketamine clearly promoted PRDM5 expression in the nucleus (Fig. 6a, b); moreover, immunostaining of brain slices further confirmed the elevated expression of PRDM5 in the nucleus of neurons (Fig. 6c, d).

We next used an electrophoretic mobility shift assay (EMSA) and ChIP-qPCR to investigate the transcriptional regulation of the *Mgll* gene. First, we predicted the binding sites of PRDM5 using JASPAR and the TRANSFAC Professional Database, and discovered two PRDM5 binding sites in the promoter of the *Mgll* gene (NCBI gene ID: 23945, 1 to −1000 bp; Supplementary Table 1). We then used EMSA to validate the specific binding of the PRDM5 protein to the *Mgll* promoter in vitro. Importantly, the PRDM5 protein was able to bind to the *Mgll* DNA nucleotide (*Mgll* probe), forming a protein–DNA complex, which shifted slower than the pure *Mgll* probe (Fig. 6e, lanes 2, 3, and 6; Fig. 6g, lanes 2, 3, and 7). The fusion PRDM5 protein also bound with the PRDM5-specific antibody to form a "supershifted" band (Fig. 6e, lanes 7, 8, and 9; Fig. 6g, lanes 9 and 10), assuring the specific binding of the PRDM5 protein to the *Mgll* promoter. We then validated the above results in vivo by ChIP-qPCR. The mice were injected with saline or ketamine (15 mg/kg, *i.p.*) for 7 consecutive days. Bilateral punches of DLS tissue from three mice were pooled together as one sample for ChIP-qPCR analysis. The effects of enzymatic digestion and enrichment efficiency are presented in Supplementary Fig. 16. Importantly, ketamine obviously enriched PRDM5 at the promoter of the *Mgll* gene (Fig. 6f, h), indicating that ketamine regulates the transcription of the *Mgll* gene through PRDM5.

**PRDM5 is required for ketamine-seeking behavior**. Because PRDM5 was upregulated in the DLS under conditions that produce hyperactivity in response to ketamine, we wondered whether PRDM5 is required for ketamine-induced behaviors. To this end, we generated a viral vector (AAV2/8) expressing either a shRNA that silences PRDM5 mRNA (AAV-sh*Prdm5*) or a scrambled shRNA control (SCR). The detailed designs of the AAVs are shown in Supplementary Fig. 17. We bilaterally injected AAV2/8 into the DLS, and after 3 weeks, compared with the scrambled control, AAV-sh*Prdm5* effectively reduced PRDM5 protein expression (Fig. 6i, j and Supplementary Fig. 18).

We next tested the effect of a bilateral DLS infusion of *AAV-Prdm5* shRNA in the self-administration and hyperlocomotion paradigms. In the ketamine self-administration paradigm, *AAV-Prdm5* shRNA significantly reduced the number of active nose pokes and drug injections (Fig. 6k, l), indicating an essential role of PRDM5 in ketamine-seeking behavior. In addition, compared to the scrambled control, AAV-*Prdm5* shRNA also elevated MAGL protein expression (Supplementary Fig. 19a, b) and reduced 2-AG levels in the DLS (Fig. 6m), suggesting the involvement of PRDM5-MAGL-2-AG signaling in the reinforcing effect of ketamine. Surprisingly, compared to the scrambled control, AAV-*Prdm5* shRNA did not reduce the hyperlocomotor activity induced by ketamine (Fig. 6n), indicating that additional or distinct target genes of PRDM5 may be involved in locomotor activity. In addition, AAV-*Prdm5* shRNA clearly elevated MAGL expression (Supplementary Fig. 19c, d) and reduced the 2-AG level in the DLS in the ketamine hyperlocomotion paradigm (Fig. 6o).

**Ketamine increases 2-AG production in primary cultured SPNs.** We further explored the roles of MAGL and PRDM5 in

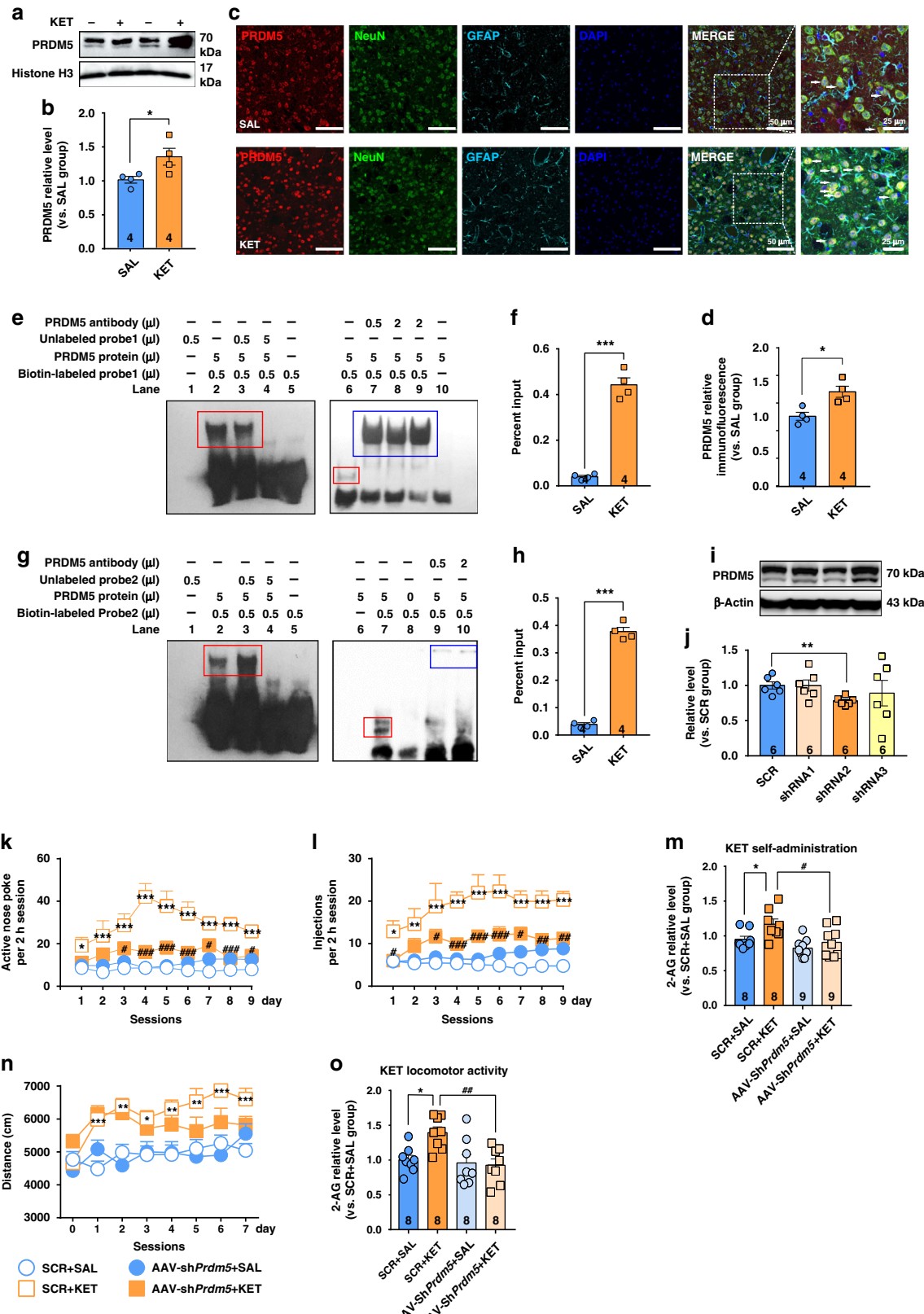

2-AG production in primary cultured SPNs. Primary SPNs began making contacts to form a neural net at day 3 in vitro (DIV3) and became closely connected at DIV7 (Supplementary Fig. 20a). Most of the isolated primary cells were SPNs, which was evidenced by the immunostaining of DARPP32 (Supplementary Fig. 20b), a marker of SPN cells[42]. DIV4 SPNs were exposed to

ketamine at a concentration of 1.5 µg/mL for 3 h per day for 4 consecutive days, followed by the measurement of MAGL and PRDM5 expression by immunoblotting. We found that ketamine significantly decreased MAGL expression, but increased PRDM5 expression in cultured SPNs (Fig. 7a, b). Moreover, the 2-AG level was obviously increased by ketamine (Fig. 7c). We then used

**Fig. 6 PRDM5 regulates *Mgll* gene transcription and ketamine-seeking behavior. a, b** Repeated ketamine exposure (15 mg/kg) increased PRDM5 protein expression in the nucleus of DLS neurons (unpaired two-tailed $t$ test, $t(6) = 2.533$, $P = 0.0445$). PRDM5 is observed as two bands at 66 and 70 kDa (two isoforms). **c** Co-immunostaining for PRDM5 (red), NeuN (green), GFAP (wathet blue), and DAPI (deep blue) in the DLS. **d** Ketamine (15 mg/kg) increased PRDM5 protein expression in DLS neurons (unpaired two-tailed $t$ test, $t(6) = 3.572$, $P = 0.0118$). **e, g** EMSA experiments were performed with 5 μg of PRDM5 protein and biotin-labeled *Mgll* DNA oligos (probe1 and probe2). The shifted protein–DNA band (red rectangle; lanes 2, 3, and 6 in **e**; lanes 2, 3, and 7 in **g** was competed with excess and unlabeled DNA oligos (lane 4). Preincubation with anti-PRDM5 antibodies resulted in supershifted antibody–protein–DNA bands (blue rectangle; lanes 7, 8, and 9 in **e**; lanes 9 and 10 in **g** that migrate more slowly in the native gel. **f, h** Ketamine (15 mg/kg) enriched PRDM5 at the promoter of gene *Mgll* (site 1 and site 2; site1, unpaired two-tailed $t$ test, $t(6) = 12.87$, $P < 0.0001$; site2, unpaired $t$ test, $t(6) = 20.32$, $P < 0.0001$). Compared to SAL group, *$P < 0.05$, ***$P < 0.001$. **i, j** shRNA2 targeting *Prdm5* effectively interfered with PRDM5 expression (unpaired two-tailed $t$ test, $t(10) = 3.908$, $P = 0.0029$). Compared to SCR group, **$P < 0.01$. **k** AAV-sh*Prdm5* reduced ketamine-seeking behaviors ($n = 21$ mice/group, two-way repeated measured ANOVA, followed by Dunnett's multiple comparisons test, virus $F(3,48) = 38.24$, $P < 0.0001$; time $F(8,384) = 2.848$, $P = 0.0044$; interaction, $F(24,384) = 2.056$, $P = 0.0027$). **l** AAV-sh*Prdm5* reduced ketamine-taking behaviors ($n = 21$ mice/group, two-way repeated measured ANOVA, followed by Dunnett's multiple comparisons test, virus $F(3,48) = 31.97$, $P < 0.0001$; time $F(8,384) = 2.554$, $P = 0.0101$; interaction, $F(24,384) = 1.016$, $P = 0.4441$). **m** AAV-sh*Prdm5* restored the DLS 2-AG level in ketamine self-administration test (one-way ANOVA, followed by Dunnett's multiple comparisons test, $F(3,30) = 5.848$, $P = 0.0029$; SCR + KET vs. SCR + SAL, $P = 0.0461$; SCR + KET vs. AAV-sh*Prdm5* + KET, $P = 0.0123$). **n** AAV-sh*Prdm5* could not reduce ketamine-induced hyperlocomotor activity ($n = 21$ mice/group, two-way repeated measured ANOVA, followed by Dunnett's multiple comparisons test: virus $F(3,80) = 7.706$, $P = 0.0001$; time $F(7,560) = 8.764$, $P < 0.001$; interaction, $F(21,560) = 2.867$, $P < 0.001$). **o** AAV-sh*Prdm5* restored the 2-AG level in the DLS in ketamine hyperlocomotion test (one-way ANOVA, followed by Dunnett's multiple comparisons test, $F(3,28) = 5.727$, $P = 0.0035$; SCR + KET vs. SCR + SAL, $P = 0.0142$; SCR + KET vs. AAV-sh*Prdm5* + KET, $P = 0.0031$). Compared to SCR + SAL group, *$P < 0.05$, **$P < 0.01$, ***$P < 0.001$; compared to SCR + KET group, #$P < 0.05$, ##$P < 0.01$, ###$P < 0.001$. Data are shown as mean ± SEM. SAL saline, KET ketamine, SCR scrambled *sh*RNA. Source data provided as a Source Data file.

immunostaining to validate these results. Consistently, PRDM5 was obviously upregulated (Fig. 7d, e), but MAGL was down-regulated in ketamine-treated SPNs (Fig. 7f, g).

We constructed a lentivirus expressing either a shRNA that silences PRDM5 mRNA (LV-sh*Prdm5*) or a scrambled SCR. The detailed design of the lentivirus is shown in Supplementary Fig. 21. SPNs were infected with LV-sh*Prdm5* for 8 h at DIV4 and exposed to 1.5 μg/mL ketamine for 3 h per day for 4 consecutive days starting at DIV7. Compared with the scrambled control, LV-sh*Prdm5* effectively reduced PRDM5 expression, elevated MAGL expression, and reduced 2-AG levels in ketamine-treated SPNs (Fig. 7h–j). Taken together, these results strongly support the in vivo findings that ketamine elevates the 2-AG level through the PRDM5-MAGL pathway.

Finally, we tested the effect of MK801 on the ECS in primary cultured SPNs. MK801 also significantly reduced MAGL expression, but promoted PRDM5 expression and 2-AG levels (Supplementary Fig. 22a–c), which was in accordance with the results from in vivo studies (Supplementary Fig. 12b–e). The similar effect of ketamine and MK801 on the ECS suggests the involvement of NMDAR in mediating the effect of ketamine on the ECS.

## Discussion
Understanding the molecular mechanisms underlying ketamine addiction will be invaluable for the development of the clinical application of ketamine and for the treatment of ketamine addiction. Here, we demonstrate that eCB signaling in the DLS plays a critical role in the psychostimulant and reinforced behaviors driven by ketamine. Ketamine decreases MAGL expression through PRDM5, leading to an increase in the 2-AG level and CB1R signaling activation (Fig. 8). Our findings shed light on the mechanism of ketamine-induced behaviors and show that eCBs may represent promising therapeutic targets for the treatment of ketamine addiction.

Lipid molecules have been identified as modulators of brain synaptic structure and are continuously synthesized and re-esterified in neurons as part of the processes of lipid remodeling and membrane biosynthesis. In the present study, we found that ketamine remodeled the lipidome in various brain regions and dramatically modulated the expression of two lipid ligands for

CB1R in the CPu. 2-AG production was significantly increased under various ketamine conditioning paradigms, whereas AEA level was increased mainly in the volitional ketamine model, but not in the hyperlocomotion model. These findings suggest that both 2-AG and AEA are required for the volitional action of ketamine; nevertheless, 2-AG appears to play a more dominant role than AEA in mediating the psychostimulant and reinforcing properties of ketamine. These findings further support the concept that different endogenous CB1R agonists exhibit distinct pharmacological profiles in the brain and mediate different neurobehavioral effects. In addition, several lipid classes, such as PC and sphingolipids, were also obviously altered by ketamine (Supplementary Data 1 and Data 2). Ketamine decreased the levels of short-chain PCs (≤34) but increased those of long-chain PCs (≥38), such as polyunsaturated fatty-acyls (PUFC-PCs). Moreover, sphingolipids, including ceramide and most hexosylceramide, were also elevated. As PCs and ceramide play important structural roles in membrane formation, the shift from short-chain PCs to long-chain PUFC-PCs may reflect an improvement of membrane fluidity[43]. Furthermore, as ceramide is a lipid second messenger involved in different neurodegenerative diseases[44,45], ketamine-upregulated ceramide suggests an alteration in the ceramide-mediated signal network. Further study is needed to elucidate the effect of these modified lipid molecules.

The MAGL overexpression mediated by AAV-*Mgll* effectively decreased the 2-AG level and blocked ketamine-induced hyperactivity and reinforcement, and these effects were markedly reversed by 2-AG supplementation in the DLS. These results indicate that MAGL functions as a critical hydrolytic enzyme in regulating 2-AG production and 2-AG signaling in response to ketamine. Under normal physiological conditions, MAGL may control the 2-AG-induced stimulation of CB1 receptors to a magnitude insufficient to produce psychostimulant effects. However, ketamine significantly elevates the DLS 2-AG level by downregulating MAGL expression, thus activating CB1R signaling and producing a rewarding effect. These findings are supported by previous studies showing that alcoholic subjects present lower MAGL activity than control subjects[46], and that the MAGL inhibitor JZL184 attenuates spontaneous withdrawal signs in morphine-dependent mice[47].

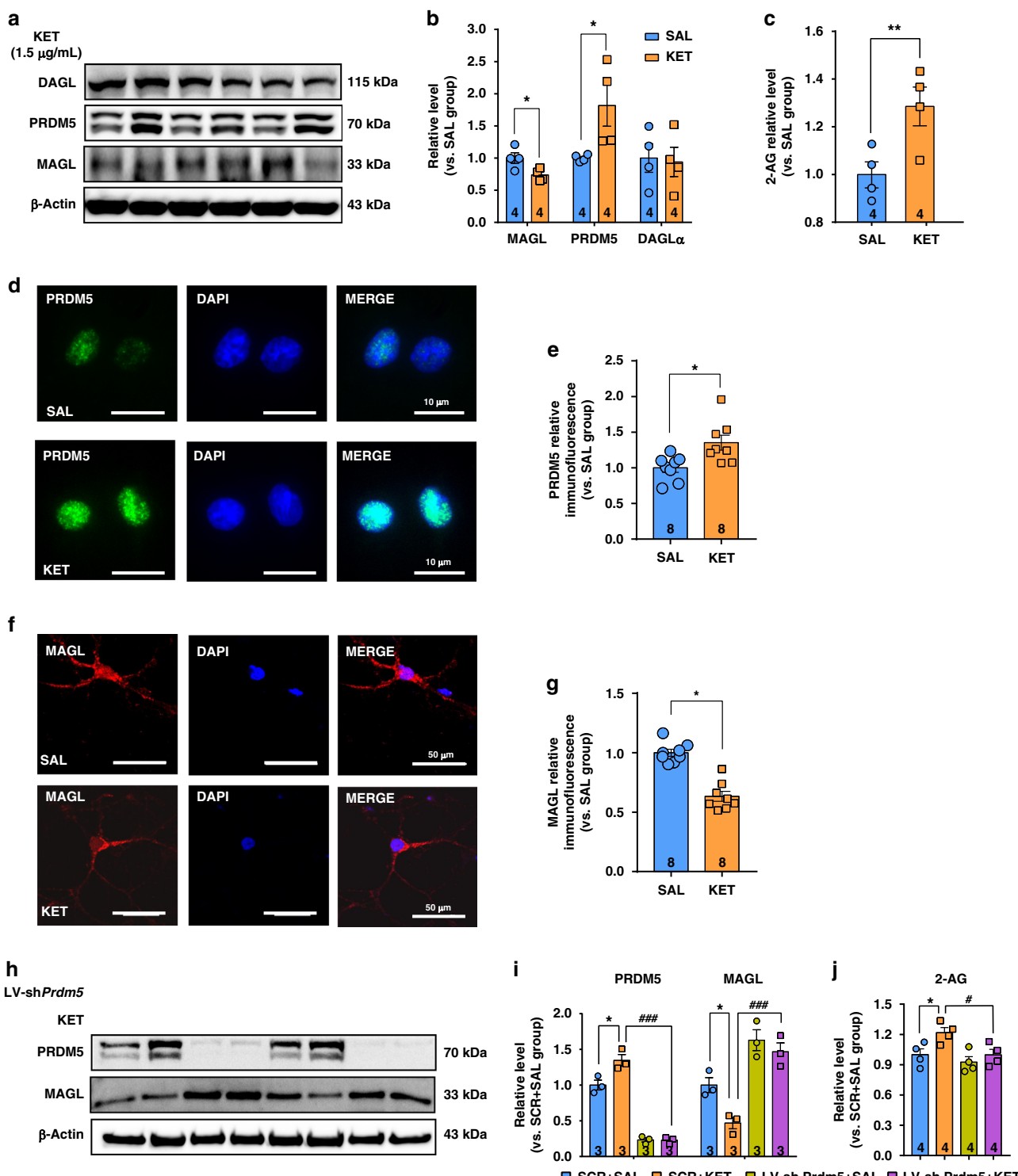

Notably, activation of the downstream molecules of 2-AG-CB1R signaling varied significantly between the ketamine hyperlocomotion and self-administration paradigms. We hypothesize that the discrepancy may be attributed to AEA because the AEA level was only increased in the ketamine self-administration paradigm. Furthermore, AEA is a potential ligand for transient receptor potential vanilloid 1, which may be involved in the drug reward effect and activate different intracellular signaling molecules[48,49]. In mice with ketamine-induced hyperlocomotion, pERK expression was upregulated, whereas pCREB expression was downregulated. In contrast, in ketamine self-administered mice, PKA was downregulated, whereas pCREB expression was upregulated. Such regulation of CREB phosphorylation seems to be inconsistent with changes in ERK or PKA activation, revealing the complexity of synaptic activity-dependent signaling involved in various ketamine-induced neurobehaviors. Although the intracellular signaling cascade for CREB phosphorylation is unclear in this study, we assume that ERK may play an important role in ketamine-induced psychomotor activity, while CREB may be important in mediating the rewarding effect of ketamine.

**Fig. 7 Ketamine increases the 2-AG level in the primary cultured SPNs. a**, **b** Ketamine decreased MAGL expression (unpaired two-tailed t test, $t(6) = 2.732$, $P = 0.0341$) but increased PRDM5 expression in the cultured SPNs (unpaired two-tailed t test, $t(6) = 2.527$, $P = 0.0449$). Ketamine did not affect DAGL expression (unpaired two-tailed t test, $t(6) = 0.1882$, $P = 0.8569$). **c** Ketamine increased the 2-AG level in the cultured SPNs (unpaired two-tailed t test, $t(6) = 2.951$, $P = 0.0256$). **d** Representative immunofluorescent images of PRDM5 (green) and DAPI (blue). **e** Ketamine increased PRDM5 expression (unpaired two-tailed t test, $t(14) = 2.89$, $P = 0.0119$). **f**, **g** Representative immunofluorescent images of MAGL (red) and DAPI (blue). Ketamine decreased MAGL expression (unpaired two-tailed t test, $t(14) = 7.16$, $P < 0.0001$). Compared to SAL group, *$P < 0.05$, **$P < 0.01$. **h**, **i** LV-sh*Prdm5* restored MAGL expression downregulated by ketamine (one-way ANOVA, followed by Dunnett's multiple comparisons test: PRDM5 $F(3,8) = 95.58$, $P < 0.0001$; VEH + KET vs. VEH + SAL, $P = 0.0071$, VEH + KET vs. LV-sh*Prdm5* + KET, $P < 0.0001$; MAGL $F(3,8) = 20.09$, $P = 0.0004$; VEH + KET vs. VEH + SAL, $P = 0.0299$, VEH + KET vs. LV-sh*Prdm5* + KET, $P = 0.0008$). **j** LV-sh*Prdm5* reversed ketamine-elevated 2-AG level (one-way ANOVA, followed by Dunnett's multiple comparisons test, $F(3,12) = 5.395$, $P = 0.0139$; VEH + KET vs. VEH + SAL, $P = 0.0386$, VEH + KET vs. LV-sh*Prdm5* + KET, $P = 0.0364$). Compared to SCR + SAL group, *$P < 0.05$; compared to SCR + KET group, #$P < 0.05$, ###$P < 0.001$. Data are shown as mean ± SEM. SAL saline, KET ketamine. Source data provided as a Source Data file.

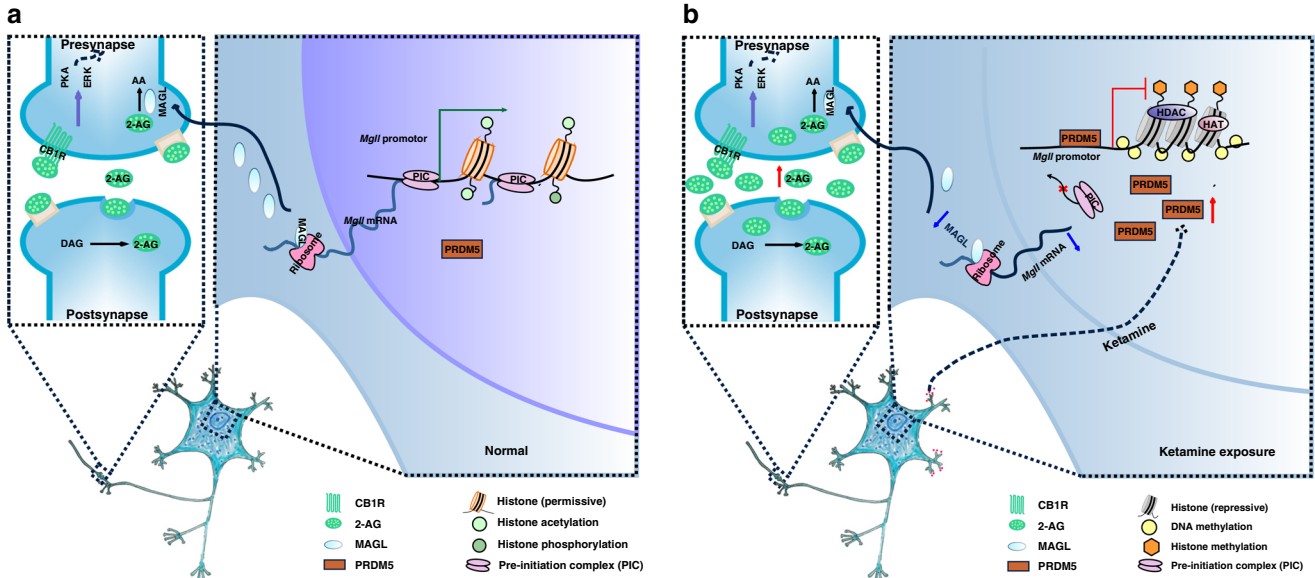

**Fig. 8 PRDM5-MAGL-2-AG-CB1R pathway in the DLS regulates ketamine addiction. a** Simplified scheme of PRDM5-MAGL-2-AG-CB1R pathway in normal mice. **b** Simplified scheme of PRDM5-MAGL-2-AG-CB1R pathway in mice under ketamine conditioning paradigm. Ketamine represses *Mgll* gene transcription by enhancing the binding of PRDM5 at the promoter of gene *Mgll*, leading to the decreased MGL expression and 2-AG hydrolysis. Consequently, increased 2-AG activates CB1R signaling, resulting in ketamine addiction-related dendritic remodeling and behaviors.

Drug-induced adaptive alterations in the structural and physiological properties of dendritic spines are crucial for long-term and compulsive drug-seeking behavior. Structural changes mainly occur in excitatory neurons in the frontal cortex and limbic regions[50–52], as well as in SPNs in the NAc[53,54]. To date, little is known regarding the dendritic complexity in the DLS upon ketamine exposure. In this study, ketamine clearly increased the dendritic complexity of SPNs in the DLS, including increasing the dendritic length, dendritic bifurcation, and spine density. This observation is consistent with previous studies showing that ketamine enhances the dendritic remodeling of cultured dopaminergic neurons in vitro[22] and promotes prefrontal spinogenesis in vivo[23]. On the other hand, intra-DLS infusion of the CB1R inverse agonist rimonabant effectively inhibited ketamine-induced dendritic remodeling, further demonstrating a critical role of DLS 2-AG-CB1R signaling in the ketamine-induced adaptive dendritic remodeling of SPNs and in persistent neurobehavioral effects. These findings provide the evidence that eCBs may selectively influence DLS-dependent memory and stimulus-dependent behaviors in the development of ketamine addiction. In addition, the expression of the α-amino-3-hydroxy-5-methyl-4-isoxazolepropionic acid (AMPA) receptor and NMDA receptor reflects, to some extent, the number of synapses[55]. We found that

the expression of phosphorylated AMPA receptor and NMDA receptor was obviously upregulated by ketamine, but downregulated by rimonabant, suggesting a remarkable capability of ketamine to increase synapse number. Spine enlargement during learning is closely associated with an increase in synaptic levels of phosphorylated (inactive) ADF/cofilin, a key regulator of F-actin dynamics and spine morphology and function[56]. We also found an increase in the expression of pCofilin, revealing a critical role of CB1R-dependent signaling in ketamine-induced dendrite remodeling.

Until now, the transcriptional regulation of the *Mgll* gene has been poorly understood. PRDM5 is a recently identified member of the PRDM family, which is composed of transcriptional regulators that modulate cellular processes and act as tumor suppressors. PRDM5 is reported to bind to both G9a and class I histone deacetylase enzymes HDAC1-3 (ref. [57]). Using an EMSA, we identified two binding sites of PRDM5 in the promoter of the *Mgll* gene and validated these sites by showing the direct binding of nucleotide probes to pure PRDM5 proteins in vitro. ChIP assays further demonstrated that ketamine rapidly increased the binding of PRDM5 to the promoter of the *Mgll* gene in vivo. Intriguingly, interfering with PRDM5 expression in the DLS effectively inhibited ketamine self-administration, but failed to

affect ketamine-induced hyperlocomotor activity, suggesting that additional or distinct PRDM5-targeted genes may contribute to ketamine-induced hyperactivity. Further work is needed to identify the genes regulated by PRDM5 under the ketamine hyperlocomotion paradigm, as these genes may represent potential targets that convey resistance to ketamine addiction. Together, our findings reveal a mechanism for the transcriptional regulation of the *Mgll* gene, through which ketamine modulates the metabolic production of 2-AG in the DLS and the magnitude of CB1R signaling.

In mesocorticolimbic reward circuits, the glutamatergic and ECSs are implicated in the neurobiological mechanisms underlying drug addiction. Previous studies have demonstrated that the ECS plays a central role in modulating glutamatergic function in cocaine-induced behavior, and in dendritic spine remodeling and synaptic plasticity[58–60]. We found that ketamine and MK801, both of which are NMDAR antagonists, exhibit similar effects on the ECS. A previous study also showed that the ECS may be involved in MK-801-induced hyperlocomotion[61]. Our results suggest that NMDAR may be involved in ketamine action on the ECS, indicating that the ECS interacts with the glutamatergic system in the mediation of the reinforcing and psychostimulant effects of ketamine.

Currently, there is no specific treatment for ketamine abuse. CB1R inhibition is capable of attenuating the addiction and relapse induced by drugs, such as cocaine, morphine, heroin, and nicotine[11,18,20,62]. However, CB1R blockage has adverse effects, such as gastrointestinal mobility and psychiatric side effects[63]. Considering that CB1Rs are the main target of 2-AG, our results suggest that the manipulation of the metabolic production of 2-AG through MAGL may be an ideal strategy for weakening the CB1R activation induced by ketamine. This strategy avoids direct and complete CB1R blockade, which may cause physical and psychological adverse effects. Manipulation of 2-AG metabolic enzymes may serve as potential therapeutic targets for the treatment of ketamine addiction, which is of particular significance for extending the clinical applications of ketamine.

## Methods

**Animals.** Male C57BL/6 J mice (6–8 weeks old) were purchased from Vital River Laboratory Animal Technology Co., Ltd. (Beijing, China). All of the mice were housed in the animal facility under a standard 12-h light/12-h dark cycle and a constant room temperature. All experimental procedures and use of the animals were in accordance with the guidelines established by the Association for Assessment and Accreditation of Laboratory Animal Care, and the Institutional Animal Care and Use Committee of Sichuan University. All the efforts were made to minimize the suffering of the mice.

**Drugs.** Ketamine HCl (abbreviated as KET in the figures) was purchased from Jiangsu Hengrui Medicine Co., Ltd. (Jiangsu, China) and dissolved in saline (abbreviated as SAL in the figures). The CB1 receptor antagonist, rimonabant (SR 141716 A, abbreviated as RIM in the figures, #S3021 Selleck) was dissolved in saline containing 6% Tween 80 (abbreviated as VEH in the figures). 2-AG (#62160, Cayman) was dissolved in saline with 2% ethanol and 2% Tween 80.

For detecting MAGL expression and 2-AG (or AEA) level, the mice were subjected to a single ketamine injection (15 mg/kg, i.p.) or repeated ketamine injections for 7 consecutive days (15 mg/kg, i.p.; once daily). Mice that received ketamine injections were sacrificed at different time after last ketamine injection. For investigating the effects of CB1R blockage on ketamine effect, mice were systemically or intracranially injected rimonanbant. For intracranial injection, mice were surgically implanted with sterilized guide cannulae in DLS. Intra-DLS injection of rimonabant (0.6 μg/μL/injection for locomotor activity test; 1 μM/injection for self-administration 0.1 μL /min) or vehicle (0.5% DMSO, 1 μL/injection, 0.1 μL/min) was performed 30 min before ketamine or saline administration.

**UPLC/Q-TOF MS/MS for untargeted lipidomic analysis.** Mice were sacrificed by cervical dislocation within 2 h after the last ketamine injection, and brain was quickly excised. PFC, NAc, dorsal striatum (CPu), Hipp, and the ACe were dissected from the brain, snap frozen in liquid nitrogen, and stored at −80 °C until assay. The liquid–liquid MTBE extraction was based on a protocol for lipidome

analysis[64]. Frozen tissue samples were spiked with internal lipids standards, 300 μL ice-cold methanol, and 450 μL MTBE. The mixture was sonicated in an ice-cold water bath for 3 min and added 250 μL 25% methanol to separate phase. Samples were centrifuged at 14,000 × $g$ at 4 °C for 15 min. The upper organic phase was collected and dried under a gentle stream of nitrogen at room temperature.

The residue was redissolved in 300 μL ACN:isopropanol:water (30:65:5). Chromatographic separation was performed using a Waters ACQUITY UPLC system (Waters Corp., Milford, USA) with a reversed phase ACQUITY UPLC HSS T3 column (2.1 mm×100 mm×1.8 μm) maintained at 55 °C by gradient elution. Mobile phase A was acetonitrile–water (40:60, v/v), and B was acetonitrile–isopropanol (10:90, v/v), both containing 10 mM ammonium acetate.

Mass spectrometric detection was performed both in positive ion mode and negative ion mode with electrospray ionization (ESI) using Xevo G2-S Q-TOF (Waters Corp., Milford, USA). Data were collected in continuum mode from 50–1200 $m/z$. Leucine-enkephalin (LE) was applied to ensure $m/z$ accuracy of the mass spectrometer, and sodium formic solution was used for the TOF mass spectrometer calibration. Data acquisition and processing were accomplished using Masslynx (Waters Corp., Milford, USA).

**Lipidomic data processing and analysis.** Progenesis QI software (Waters) was used to analyze the lipidomic data acquired from UPLC/Q-TOF MS, and the lipids were identified from Lipid Maps Database (www.lipidmaps.org) and the Human Metabolome Database (http://www.hmdb.ca/) according to their MS characteristics. Data sheets from Progenesis QI software were obtained and absolute intensities of all identified compounds (normalized abundance) were recalculated to relative abundances of lipid molecules (vs. saline group). The details of normalization method could be found at the website (http://www.nonlinear.com/progenesis/qi/v2.3/faq/how-normalisation-works.aspx). Pareto scaling was used for final statistical models. The data were processed by supervised partial least-squares discriminate analysis methods to obtain group clusters. Lipid molecules with the highest impact on the group clustering were identified in the variable importance (VIP) plots (VIP > 1). An optimized false discovery rate approach was applied to control the false positive.

**UPLC/Q-TOF MS/MS for eCB detection.** Brain tissue was prepared by using an ethyl acetate/hexane (9:1, v/v) extraction adapted from a previously published method with modification[65,66]. Tissues were spike with ethyl acetate/hexane and ISs (d4-AEA, at 10 μg/mL; d5-2-AG at 100 μg/mL, 2 μL mixed IS for each sample), and ultrasonicated in an ice-cold water bath. Then the mixture was washed with ~10% of its volume of water and transferred to a centrifugation tube. Samples were centrifuged at 7000 × $g$ at 4 °C for 15 min, and the supernatants were transferred to clean tubes. The homogenization and centrifugation steps were repeated three times, with the supernatants pooled to optimize eCBs recovery. Then the pooled supernatants were evaporated to dryness under a gentle stream of nitrogen.

The dry residue was dissolved in 100 μL acetonitrile, and 5 μL reconstituted extract was injected into the LC/MS–MS system for analysis. Chromatographic separation was performed using a Waters ACQUITY UPLC system (Waters Corp., Milford, USA) with a reversed phase ACQUITY UPLC HSS T3 column (2.1 mm × 100 mm × 1.8 μm) maintained at 40 °C by gradient elution with a mobile phase flow rate of 0.3 mL/min. Gradient elution mobile phases consisted of A (acetonitrile with 0.1% formic acid) and B (water with 0.1% formic acid).

MS/MS analyses were accomplished in positive ion mode with ESI using Xevo G2-S Q-TOF (Waters Corp., Milford, USA). For targeted MS/MS scan, cone voltage was set as 30 V for AEA and d4-AEA, 10 V for 2-AG, and d5-2-AG. LE was applied to ensure $m/z$ accuracy of the mass spectrometer, and sodium formic solution was used for the TOF mass spectrometer calibration. A comparison of the paired ion (precursor and product ion $m/z$ values) and LC retention times with standards served to confirm the identification of eCB. An ion pair was 348.2903/131.0946 for AEA, 379.2848/361.2748 for 2-AG, 352.3154/135.1197 for AEA-d4, 384.3162/366.3057 for 2-AG-d5. 2-AG is chemically unstable in aqueous solutions and it is likely to form 1-AG by molecular rearrangement. We thus calculated 2-AG by summing up 2-AG and 1-AG. Data acquisition and processing were accomplished using Masslynx (Waters Corp).

**Intra-DLS delivery of adeno-associated virus serotype 2/8.** Under pentobarbital sodium (80 mg/kg) anesthesia, mice were positioned in a small animal stereotaxic instrument, and the cranial surface was exposed. Virus was delivered bilaterally using Hamilton syringes for a total of 0.5 μL of virus into the DLS at a 10° angle [anteroposterior (AP), +0.6 mm; mediolateral (ML), ±2.60 mm; dorsoventral (DV), 3.5 mm] at a rate of 0.1 μL/min. Animals receiving AAV injections were allowed to recover for 3 weeks before beginning behavioral tests. 2-AG-treated mice were recovered for 2 weeks before cannulation, and then continued to recover for 1 week after cannulation.

**Intra-DLS cannulation.** Under general pentobarbital sodium (80 mg/kg) anesthesia, mice were bilaterally implanted with a sterilized catheter (27 gauge stainless steel, 3.5 mm projection). Each catheter was positioned on the skull just on the DLS (AP, +0.6 mm; ML, ±2.60 mm; DV, 3.5 mm). Mice were allowed 7 days to recover

from surgery before the behavioral tests. The implanted catheters were used for rimonabant or 2-AG delivery.

**Intravenous catheterization**. Mice were anesthetized with pentobarbital sodium (80 mg/kg) and prepared with long-term catheters (internal diameter, 0.40 mm; external diameter, 0.48 mm), as modified from a previous study[67]. Briefly, the catheter was inserted into the right jugularvein and passed subcutaneously over the right shoulder to exit dorsally between the scapulae. Animals were given 7 days to recover from surgery before starting self-administration tests. During the first 4 days after surgery, animals received daily antibiotic treatment (penicillin, 160,000 U/mL, i.m.; 0.1 mL/mice), and catheters were flushed with 0.04–0.06 mL of heparanized saline (30 U/mL 0.9% sterile saline) to prevent coagulation.

**Recombinant adeno-associated virus and lentivirus**. The plasmid for *Mgll* (NM_011844) expression was designed as AAV-*Mgll* (pAAV-CAG-EGFP-2A-*Mgll*-3FLAG), and the control plasmid designed as AAV-eGFP (pAAV-CAG-EGFP-2A-MCS-3FLAG). These two vectors contained an enhanced GFP coding sequence, allowing the identification of infected cells. All vector insertions were confirmed by dideoxy sequencing. The detailed constructions of the plasmids are presented in Supplementary Fig. 13.

Three shRNA sequences targeting mice *Prdm5* were designed to guarantee the interfering effect on *Prdm5*. The pAAV-hU6-3FLAG vector was used to express Prdm5 (NM_027547) interference sequence. The negative control (NC) vector expressed scrambled shRNA, which was generally applied as the control and theoretically had no effect on any gene. The plasmids design and construction of *Prdm5* shRNA and scrambled shRNA are shown in Supplementary Fig. 13. Through pretest, #2 shRNA (target GCCCATATTGTGGCCAGAA) sequence was chosen for the following experiments for its significant interfering efficiency (Fig. 6i, j). Virus titers were between $10^{12}$ and $10^{13}$ genomic copies per mL for all batches of virus used in the study.

Three shRNA sequences targeting rat *Prdm5* were designed to guarantee the interfering effect on *Prdm5* in cultured rat striatal neurons. The pLKD-CMV-eGFP-U6-shRNA vector was used to express Prdm5 (Rat Genome Database, XM_017602810.1) interference sequence. The NC vector expressed scrambled shRNA, which was generally applied as the control and theoretically had no interference on any genes. The plasmids design and construction of *Prdm5* shRNA and scrambled shRNA are shown in Supplementary Fig. 21. Virus titers were $4 \times 10^{8}$–$7 \times 10^{8}$ genomic copies per mL for all batches of virus used in the study.

**Primary cultured striatal neurons and virus transfection**. Primary striatal neurons were prepared from Sprague–Dawley rat at embryonic day 19 using previously described methodology with some modification[68]. The striatal neurons were plated at a density of $1.0 \times 10^{7}$ cells/10 cm$^2$ dish precoated with poly-D-lysine or 100,000 cells/well on cover glass precoated with laminin and poly-D-lysine on 24-well plates. Striatal neurons were plated in cold Neurobasal Medium (#21103049, Thermo Fisher) with horse serum (#16050122, Gibco) and glutamine (#25030081, Gibco) and kept at 37 °C. After 6 h of incubation, culture media was changed to neurobasal medium with B27 supplement (#0080085SA, Gibco).

The neurons (at DIV4) were exposed to ketamine (1.5 μg/mL) or saline 3 h per day, for 4 consecutive days. After the last exposure of ketamine at DIV7, neurons were collected for 2-AG detection, immunoblotting, and immunofluorescence. For lentivirus transfection, the neurons were transfected with LV-*sh*Prdm5 (multiplicity of infection=10) at DIV4. After 8 h of incubation with lentivirus, culture media was replaced with fresh medium. Seventy-two hours after transfection (at DIV7), the neurons were exposed to ketamine (1.5 μg/mL), as described above and collected for assays.

**Locomotor activity**. Locomotor activity sessions were conducted once daily. Each mouse was placed in a chamber, followed by i.p. injection of 15 mg/kg ketamine or saline. Locomotor activity was measured for 15 min per day, as previously described[69]. The chambers were black acrylic boxes ($48 \times 48 \times 31$ cm) that were equipped with a top unit including a camera. Automated tracking was performed using EthoVision version 7.0 software (Noldus Information Technology). Animals were reweighed each day prior to behavior. After locomotor activity assay, the mice were sacrificed and brain tissue was collected for eCB detection, cAMP level measurement, immunoblotting, immunohistochemistry, ChIP-qPCR, as well as Golgi staining.

**Ketamine self-administration**. Ketamine self-administration was performed as previously described with modification[70]. Sterilized catheters were surgically inserted into the right jugular vein of mice. Animals were given 7 days to recover from surgery. The mice had 2 h of access daily to ketamine (0.5 mg/kg/infusion) under a FR1 reinforcement schedule. The data were collected by Anilab 6.40 Software. The mice in this paradigm were used for lipid measurement by HPLC–MS/MS and immunoblotting.

**Dose response and sensitivity to ketamine in self-administration model**. The test was performed as previously described with modification[41]. Sterilized catheters were surgically inserted into the right jugular vein of mice; meanwhile, sterilized

catheters were implanted in the DLS for 2-AG infusion. Self-administration training sessions lasted 3 h/day for 5 day/week over 3 weeks (15 sessions) until criteria for stable responding was obtained (<15% variance over the final 3 days of self-administration training). The FR requirement was incrementally increased first to FR3 and then to FR5 by the end of a training period of 3 weeks. Following cocaine self-administration training and stabilization, mice received dose–response on FR self-administration tests, as described and outlined in Fig. 5m. Mice were tested in a within-session cocaine self-administration dose–response test before and after 2-AG infusion. Each within-session dose–response test began with a 30 min loading phase with access to the training dose (0.5 mg/kg/injection), followed by five consecutive 30 min components when descending injections doses were available (1.0, 0.3, 0.1, 0.03, and 0 mg/kg, i.v.). Dose–response data of ketamine self-administration were averaged across 2 days. Baseline measures in the 2-AG-supplemented mice were counterbalanced based on baseline self-administration and dose-dependent response.

**Measurement of MAGL activity**. According to the manufacture instruction, MAGL activity in CPu was measured by using the MAGL inhibitor screening assay kit (#705192, Cayman Chemical). Briefly, MAGL hydrolyzes 4-nitrophenyllacetate, resulting in a yellow product, 4-nitrophenol, with an absorbance of 405–412 nm. The absorbance of 405–412 nm represents the amount of product and the activity of MAGL. JZL195 included in the assay kit was used as positive control.

**RNA extraction and qRT-PCR assay**. RNA was isolated from brain sample using TRIzol reagent (Life Technologies, USA) according to the manufacturer instructions. The extraction was purified with RNeasy Micro Columns, and spectroscopy confirmed that the RNA 260/280 ratio was 1.8–2.0. RNA was reverse transcribed using Bestar™ qPCR RT Kit (DBI-2220) and quantified by qPCR using Bestar® Sybr Green qPCR mastermix (DBI-2043). Each reaction was run in triplicate and analyzed following the Ct method, as previously described[71]. All primer sequences are listed in Supplementary Table 2.

**Protein extraction and immunoblotting**. Brain tissues and SPN neurons were extracted using protein extraction kit containing protease inhibitor cocktail and DTT (#K269, Biovision, Milpitas, CA). Phosphatase inhibitors (Roche, Germany) was added into the extraction to prevent the degradation of phosphorylated proteins. Protein concentration was determined by a commercial Bradford kit (Beyotime, Jiangsu, China).

Equal amounts of protein (20 μg) were loaded and separated by 10% sodium dodecyl sulfate–polyacrylamide gel (Beyotime, Jiangsu, China), and then transferred to polyvinylidenedifluoride membrane (Millipore, Billerica, MA, USA). After blocking with 5% nonfat milk in TBST for 1 h at room temperature, the membrane was probed with primary antibodies overnight at 4 °C and then was incubated with secondary HRP-conjugated antibodies for 2 h at room temperature. After three more washes, blots were coated with ECL 2 (Thermo Pierce) and exposed to ChemiScope Mini (Qingxiang, Shanghai, China). Phosphorated proteins and total proteins were analyzed with Chemi Analysis software. Immunoblotting experiments were repeated at least twice to ensure reproducibility. The detailed information of the primary and secondary antibodies is shown in Supplementary Data 3.

**Immunohistochemistry**. Mice were sedated with pentobarbital sodium (80 mg/kg, i.p.) and perfused with 1× PBS followed by 4% paraformaldehyde (PFA). Brains were postfixed overnight in 4% PFA then were cryoprotected overnight in 30% sucrose in PBS. Brains were sliced on a microtome at 10 μm. The slices were permeabilized and blocked in 5% BSA, 0.3% Triton X-100, 0.2% Tween-20, 10% normal donkey serum in PBS, then incubated with primary antibodies (Supplementary Data 3) in blocking buffer at room temperature for 2–4 h or at 4 °C for overnight. Following a series of 1× PBS rinses, slices were incubated for 1 h at room temperature with secondary antibodies while protected from light. Slices were counterstained with DAPI, mounted and cover slipped with Prolong Gold anti-fade mounting medium. Images were acquired on an A1RMP + two-photon confocal microscope (Nikon) under the same conditions to compare intensities. Immunohistochemistry experiments were repeated at least triplicate to ensure reproducibility. Images were analyzed by Image J software.

Striatal SPN neurons (E19 rat) were fixed with 4% PFA in 1× PBS for 20 min at room temperature. The following procedures were the same as brain slices staining. The primary and secondary antibodies were listed in Supplementary Data 3. Immunohistochemistry experiments were repeated at least triplicate to ensure reproducibility. Images were analyzed by Image J software.

**cAMP concentration measurement**. According to the manufacture's protocol, cAMP concentration of brain tissue was quantitatively measured by the commercial kit (#ADI-900-067, Farmingdale, NY, USA). Briefly, the frozen tissue (CPu, ~5 mg) was weighed, added 250 μL 0.1 M HCl, and homogenized on ice using a Polytron-type homogenizer. The samples were centrifuged at $16000 \times g$ for 5 min. After dilution with PBS at a ratio of 1:10, the supernatant sample was used for detection of CPu cAMP level. The protein concentration in the supernatant was also determined. The cAMP concentration was normalized by protein concentration and recalculated the relative level to the vehicle–saline group.

**Golgi–cox staining**. Animals were anesthetized with sodium pentobarbital (80 mg/kg, i.p.) before euthanizing. Brains were removed quickly from the skull to avoid any damage. After rinsing, tissue was sliced in ~10 mm thick blocks. The blocks were stained with the FD Rapid Golgi Stain[TM] kit (FD Neuro Technologies, USA). Briefly, the blocks were first immersed in the impregnation solution (A and B), which was replaced after 12 h and then kept in dark for 14 days. Afterward, the blocks were put in solution C, which was replaced after 24 h and kept in dark for the next 48–60 h. Then the blocks were sliced on a cryomicrotome at 100 μm. Slices were mounted on a gelatin-coated microscope slides, stained, and dehydrated and cover slipped with Permount.

**Dendritic spine analysis**. Dendrites were acquired under an Olympus confocal microscope. Dendritic complexity analysis was carried out using the Image J software and Sholl Analysis plugin. SPNs included in the analysis satisfied the following criteria: (i) an isolated cell body with a clear relationship of the primary dendrite to the soma, (ii) a presence of untruncated dendrites, (iii) consistent and dark impregnation along the extent of all of the dendrites, and (iv) little overlap with the neighboring impregnated cells[72]. For each included neuron, cell body and dendritic branches were reconstructed under 40× magnification (NA = 0.9). Concentric ring intersections were determined using the Image J sholl analysis plugin at 10 μm increments from soma[37] (Supplementary Fig. 8b). Spine density was measured under a 100× oil lens (NA = 1.4) with Image J software (Supplementary Fig. 8c).

**Electrophoretic mobility shift assay**. PRDM5 protein was expressed in 293 T cells and then was extracted. The protein concentration was adjusted to 1 μg/μL. An EMSA was performed using a LightShift[TM] Chemiluminescent EMSA Kit (ThermoFisher) according to the manufacturer's protocol. The probe sense sequences used for EMSA were as follows: wild-type probe1, 5′- AATGCGCG GTGCCGCGGAGCGCGTCTCGCAGCAGCTCCGGG-3′, and wild-type probe 2, 5′-CCTGCAGCCCAGGCGGGGAGGGCGCGGACCCCGTGGTGCTGCCC-3′, both of which corresponded to the *Mgll* promoter. The wild-type *Mgll* probe was labeled with biotin. The PRDM5 protein (5 μg) was incubated with 1 μg poly [d(I-C)], the binding buffer included in the kit and biotin-labeled wild-type probe in the presence or absence of an unlabeled probe for 15 min at room temperature. The bound DNA complexes were separated by 5% nondenaturing polyacrylamide gel electrophoresis and transferred to a nylon membrane (ThermoFisher). The nylon membranes were cross-linked, and chemiluminescent detection was performed using streptavidin–horseradish peroxidase conjugate and the chemiluminescent substrate. The signals were recorded using the ChemiDoc MP (Biorad).

For the supershift analyses, PRDM5 antibody (0.5 or 2 μg, SantaCruz) was added to the nuclear extracts in gel shift buffer (above) for 1 h at 4 °C, followed by the addition of the probe.

**Chromatin immunoprecipitation-qPCR**. ChIP was carried out with an enzymatic chromatin IP Kit (Cell signaling). The cross-linked chromatin from bilateral punches of DLS (tissues from three mice were pooled together as one sample) was digested into an average of ~150 bp fragment by micrococcus and then was sonicated to release from nuclear, then immunoprecipitated with specific PRDM5 antibody (5 μg, ChIP grade, #sc-376277x, Santa Cruz) or an IgG control. After being precipitated with protein A beads, the beads were washed with low salt, high salt, and LiCl buffers to remove nonspecific DNA binding. Eluted chromatin was reverse-cross-linked at 65 °C in the presence of proteinase K and EDTA. DNA was purified and the enrichment of specific promoters was measured directly via real-time qPCR (DBI-2043). The primers for ChIP are shown in the Supplementary Table 3. ChIP data were shown as the percentage of the input.

**Preparation for striatum single-cell suspension and flow cytometry assay**. Brain tissue was dissected on ice and enzymatically digested using Adult Brain Dissociation kit (#130-107-677, Miltenyi) according to the manufacturer's procedure. Single-cell suspension was prepared for flow cytometry assay. Single-cell suspensions from CPu were centrifuged and the pellet was resuspended with a viability dye Fixable Viability Stain 780 (1:1000; BD) for live or dead cell discrimination, followed by incubation with blocking buffer containing CD16/CD32 (1:100; BD) for 10 min. Cell suspensions were then incubated with Foxp3/Transcription Factor Staining Buffer Set (#00-5523, Invitrogen) and the corresponding antibodies at 4 °C for 30 min in the dark. The samples were washed, centrifuged, and resuspended in staining buffer containing 2% fetal bovine serum. Cells were then sorted using a FACSAria SORP (BD, USA). Gating parameters and data analysis were performed using FlowJo 10 software (Tree Star, USA).

**Statistics and reproducibility**. Unless otherwise noted, statistical analysis was done using unpaired Student's *t* test (two tailed) and two-way ANOVA. Data from locomotor activity paradigm, active nose pokes and drug injections in ketamine self-administration paradigm was analyzed by two-way repeated measured ANOVA, post hoc analysis was done using Tukey's test, Dunnett's test, or Bonferroni's test with Graphpad Prism version 7. One-way ANOVA was performed to analyze specific lipids, gene and protein expressions, and Dunnett's *t* test or Tukey's test was performed for post hoc analysis by using Graphpad Prism version

7. Minimum level of significance was set at $P < 0.05$. Data are presented as mean ± SEM. All experiments were carried out one to three times, and data replication was observed in instances of repeated experiments. Details on each statistical test can be found in the Source Data file.

**Reporting summary**. Further information on research design is available in the Nature Research Reporting Summary linked to this article.

## Data availability

All data and statistics reported in the main text can be found in the Supplementary Data files. Source data are provided with this paper.

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

## Acknowledgements

This work was partially supported by the National Natural Science Foundation of China (Grants 81871043, 81571301, and 81401105), National Science and Technology Major Project (2018ZX09201017 and 2018ZX09201018), and 1·3·5 Project for Disciplines of Excellence, West China Hospital, Sichuan University (ZYGD18024).

## Author contributions

W.X. and H.L. performed the main behavioral tests, Golgi–Cox detection, immuno-blotting, immunohistochemistry, ChIP-qPCR, EMSA test, and primary cultured SPNs. L. W. and C.L. performed some primary cultured SPNs. J.Z. and X.W. performed some behavioral tests and LC–MS detection. Y.H. and Q.F. prepared frozen brain slices. Y.X. assisted with LC–MS detection. X.L., Q.B., H.W., J.T., and Y.Z. helped designing experiments and provided experimental reagents. W.X. and X.C. designed all experiments, interpreted the results, and wrote the paper.

## Competing interests

The authors declare no competing interests.
