## [Peer Review File · Nature Communications]

Reviewers' comments:

Reviewer #1 (Remarks to the Author):

Nature Communications Review, NCOMMS-19-32869-T, 2019 Nov

Xu et al.: Endocannabinoid Signaling Regulates the Reinforcing and Psychostimulant Effects of Ketamine

In this study, Xu and collaborators provide strong data regarding the impact of ketamine on the endocannabinoid system in a mouse model. Using a combination of lipidomic, cell morphological, pharmacological, in vivo gene manipulation and behavioral tools, the authors demonstrate that endocannabinoids, and particularly 2-AG, play a critical role in the psychostimulant and reinforcing effects of ketamine, especially in the caudate-putamen (CPu). The data also show that MAGL, the hydrolytic enzyme of 2-AG, is regulated by negative transcriptional factor PRDM5, revealing a potential mechanism for ketamine-seeking behavior. Overall, the study provides important information about the neurobiological effects of ketamine, a drug used in the clinic for different applications but also abused in a non-medical environment. In addition, the data highlight the endocannabinoid system as a potential novel therapeutic target for ketamine addiction that supports published data that has demonstrated a significant effect of ketamine and glutamatergic regulation on the endocannabinoid system. The manuscript is presented well and the experimental approaches as well as the results are solid. The manuscript could be improved by addressing the following suggestions:

1. Overall, the data is strong but the literature is not well cited since it has already been shown that ketamine alters the endocannabinoid system and strongly established of the glutamatergic regulation (both metabotropic and NMDA) of endocannabinoid levels. Such findings need to be fully acknowledged.
2. The Abstract states that CPu 2-AG production may be a potential novel therapeutic target for ketamine addiction. How can CPu PRDM5 truly be a rational therapeutic target? It would be best to rephrase this sentence and other similar lines in the paper.
3. The last paragraph of the Introduction mentions genes that should be described in a bit more detail for the general audience at the start: please define that Mgl1 is the gene encoding for MAGL and the known role of PRDM5.
4. PRDM5 was chosen to be one focus of the study (experiments in Figs. 6-8) but there is no rationalization as to why this particular transcription factor was studied considering all the various transcriptional regulatory mechanisms that are expected to play a role in regulating Mgl1. Please explain the rationale in the context of specific functions.
5. No rationale is given as to why the 15 mg/kg ketamine dose was chosen for the experiments. It would be important to justify, e.g. that it is the equivalent of ketamine commonly used recreationally. Also, what was the dosing schedule in the morphological experiments presented in Fig. 3?
6. Table 1 shows a lot of negative changes especially in the CPu. Please provide a rationale as to why the focus of the rest of the paper is on increased endocannabinoid levels.
7. It is difficult to read the names of the different lipids presented in Fig. 1c. For simplicity, this could be moved to the Supplement. The results shown in Fig 1i and 1k seem to be switched in how they are described in the text on p7, since it refers to Fig. 1k for a single ketamine dose and Fig. 1i for repeated administration when the figure shows the opposite.

8. The focus on 2-AG instead of AEA to ketamine addiction is based on the significant 2-AG alterations observed in the hyperlocomotion and self-administration models. However, in Fig. 1f, the lack of significant effect of ketamine to increase AEA is actually due to very large variability in the saline animals that is not evident in other brain areas. It raises questions about technical issues especially since there was no such variability in saline animals in Fig. 1h where significant AEA elevation in the CPU is induced by ketamine. Given the link of glutamatergic transmission with AEA, one also wonders whether levels of AEA-related enzymes such as FAAH are altered which could provide fundamental knowledge about the endocannabinoid pathway(s) relevant to ketamine.

9. In Fig.2b, rimonabant seems to have an effect on hyperactivity starting at the fifth session only. Is there any explanation for this delay?

10. The causal data regarding MgII knockdown is very strong. Is there also evidence on the individual level that ketamine disrupts the normal relationship between the behavior and molecular regulator for the genes and proteins that were studied in the animals in Figs. 1e-1h and 2h-2k?

11. Fig.4j shows that MAGL expression is still downregulated at 72h, whereas the levels of 2-AG are back to baseline at the same timepoint (Fig.1k). In this case, restored 2-AG level would not be consistent with MAGL downregulation. Is there any compensatory mechanism arising? While not extremely important for the overall conclusions of the study, the levels of enzymes involved in 2-AG biosynthesis could be measured at the same timepoint. Please at least provide a possible explanation.

12. There is a confusion around the symbols for the different treatments in Fig.S4c. Please use the correct shapes and colors for the definitions.

13. Lines 356-364 (information on Fig. 5d-l) show experimental validation and might be more logical to describe before the other tests that use these approaches to address behavioral effects.

14. It is stated that infusion into the dorsolateral striatum with 2-AG shifted the ketamine dose response curve. However, the effect is rather small and only observed at one dose. This and other results in the figures highlight the fact that no F-values or other outcome from statistical models are provided in the Result section.

15. In Fig.6j, are these all shRNA instead of si?

16. In Fig. 6n and o, adding "2-AG" in the title of the y axis would make it easier to understand what was done.

17. Fig. 8 shows many epigenetic mechanisms but none of them were examined in the study, so it is excessive and not very informative. To make the summary more meaningful, perhaps show two simpler versions: the normal regulation vs. ketamine effects next to each-other. It would be more visually logical to show pre-synapse on the top and post-synapse on the bottom. Moreover, there is not sufficient published evidence for 2-AG binding to AMT, so that could be deleted.

18. It is stated in the Discussion, line 450, that ketamine increases MAGL, when it actually reduced it.

19. Lines 488-493: it is an over-speculation that there is a dose-dependent effect of 2-AG on a specific cell-type specific as there is not sufficient data showing this and even the potential "shift" of CB1R activation between glutamatergic and GABAergic terminals.

20. It would be helpful to provide information about the drug doses used in the figure legends.

Reviewer #2 (Remarks to the Author):

The authors examined roles of endocannabinoids in the psychoactive effects of ketamine using a combination of lipidomic, biochemical, molecular, morphological and behavioral approaches to assess endocannabinoid levels and their roles in ketamine locomotor activation and self-administration. The self-administration approach is especially powerful as it indicates mechanisms that might be involved in abuse of the drug. The authors also explored the role of factors that control expression of endocannabinoid metabolizing enzymes. Effects of chronic versus acute exposure were also elucidated. Overall, the manuscript presents new and intriguing findings linking endocannabinoids to ketamine actions. However, there are significant questions about the link between ketamine and changes in endocannabinoid function, as well as concerns about some of the experimental approaches, reagents and interpretations. In addition, the manuscript needs significant revision to clarify several points.

Major Comments

- 1) It is not clear if the ketamine interactions with the endocannabinoid system are secondary to initial actions at the NMDA receptor or another molecular target. Given the current controversies about the locus of initial ketamine and ketamine metabolite effects on the brain it would be good to address this question.
- 2) The rationale for examining CB1 coimmunoprecipitation with Galpha/i is not clear. In general, activation of GPCRs should decrease, not increase, interaction with Galpha subunits, and this occurs on a very rapid timescale that may not be captured by a technique with temporal resolution like Co-I. Why rimonabant would block this increased Co-I is also unclear, as it should, if anything, promote increased affinity for Galpha/i by itself. Overall, it seems unclear that the authors know how GPCRs function.
- 3) In the initial lipidomics screen the data need to be corrected for multiple comparisons and/or false discovery rate needs to be calculated.
- 4) The locomotor effects of some treatments appear to be only in the later test sessions (e.g. figures 2b and 6k), while in other experiments the effects are there from the beginning of treatment. Can the authors explain this apparent discrepancy?
- 5) Sample sizes appear to be very small in some experiments (e.g. figure 1h-i). Did the authors perform a power analysis to determine what would be needed to support or reject the null hypothesis?
- 6) Figures 4d and 7f, why is MAGL so strongly expressed in somata and dendrites? In most brain regions this enzyme is mainly in presynaptic terminals. How reliable is the MAGL antibody?
- 7) In the pERK immunoblots only the lower molecular weight form is increased. It is not clear what this should happen, but it might be due to insufficient substrate in the assay.
- 8) What was the source of the MAGL used in the enzyme activity assay?
- 9) Was expression of virally-delivered constructs specific for neurons?
- 10) Figure 6c, is the red labeling supposed to be visible in the MERGE images? If so, it is not.
- 11) Why was a CMV promoter used for the shRNA construct? In general, this promoter is not that strong in neurons in situ, and may also drive expression in glia.

12) The authors need to provide information on the specificity of all the antibodies used in the study. Preferably, specificity should be shown using knockout mouse tissue. Also, were protease inhibitors used in the tissue preparation and blotting procedures (they should be). What were the secondary antibodies for the striatal SPN culture experiments?

13) In the ChIP-qPCR assay, how much of the material from the punches comes from SPNs, and could there be a signal-to-noise ratio problem due to the use of this approach?

Minor Comments:

a. Page 3, the statement that "...the precise mechanisms underlying the psychopharmacological effect of ketamine is largely unknown." is an overstatement. Several primary targets for ketamine action are known, the NMDAR being the most prominent one. Downstream effects may not be as clear, but at least we know a great deal about initial actions in the brain.

b. The last paragraph of the introduction may confuse readers. A brief description of PRDM5 and the rationale for examining this should be provided earlier in this section to provide the context for the last sentence.

c. Several potentially interesting changes in lipid expression, including some decreases, are noted in table 1 (provided they survive proper analysis). Clearly, these changes cannot be examined in any detail in this study, but they should at least be noted in the discussion.

d. For the locomotor activity analysis were animals always examined in the same environment, or was a novel environment used each time?

e. What tissue fixation, if any, was used in the Golgi-Cox procedure?

f. Why is PRDM5 data presented in figure 4e when this protein has not yet been discussed? Is this graph mislabeled?

g. Figures 6 and 7, the y-axis label "Relative Level to Internal Standard" is uninformative. Please indicate what was actually measured (e.g. 2-AG in figure 6n).

h. Figure 7b, should the units be μM rather than μmol ?

i. Figure S17b, at what DIV were these neurons examined. The DAPI staining seems to indicate pretty sparse neuronal expression.

j. Page 7, end of first paragraph, at this point the data only show a correlation with increased endocannabinoid expression, so this last sentence should be reworded.

k. Rimonabant should be referred to as an inverse agonist.

l. Page 8, first paragraph, make it clear that the PKA and CREB assays were performed on tissue from animals that self-administered ketamine, thus the effect may be due to the combination of the drug and the learning itself.

m. Page 9 and 10, the authors should first describe the ketamine-induced changes in dendritic arborization before describing the rimonabant block.

n. While the term "clastic" is technically correct for description of these enzymes, it is a rather old-fashioned term. Perhaps catabolic, degradative, or another term would be better.

o. Page 16 last paragraph, make it clear that the nuclear labeling was used for PDRM5 localization

while dendritic localization with MAP2 was used to examine MAGL subcellular location.

p. Supplement, line 56, I assume the authors mean to say ketamine and not cocaine.

q. Supplement, line 162, the authors indicate that only 1 week was given between AAV injections and behavioral testing onset. Is this time period sufficient for viral infection and expression?

r. There are several grammar and usage errors in the manuscript, especially in the supplemental material. Additional editing is required. For example, there is no "Turkey's" post hoc test.

Reviewer #3 (Remarks to the Author):

This study examines the role of 2-AG in the locomotor activating and self-administration effects of ketamine. The results demonstrate that repeated ketamine administration or self-administration increases levels of 2-AG in the dorsal lateral striatum, and that this effect is mediated by decreased expression of the metabolizing enzyme MAGL. Further, the results show that the decrease in MAGL is mediated by increased levels of the transcription factor PRDM5, a negative regulator of MAGL gene expression. The studies include evidence that overexpression of MAGL or knockdown of PRDM5 via viral expression of shRNA block the effects of ketamine on locomotor activity or self-administration, although with differential effects depending on which is targeted. In general, these studies are well designed and the results provide evidence that regulation of 2-AG, MAGL, and PRDM5 are involved in the actions of ketamine, although there are several points to address.

1. The results section states that levels of 2-AG were significantly increased at 6, 24, and 48 h after the last ketamine injection, but this is not consistent with what is shown in Fig. 1i and Fig 1k? This should be addressed.
2. The rimonabant studies are confusing as higher systemic doses (1-3 mg/kg, ip) decrease but lower doses (0.1 and 0.3 mg/kg) increase locomotor activity. This is in contrast to what is stated in the text of the results. In addition, the dose of 0.6 mg/kg that blocks the locomotor effects of ketamine also decreases activity. Also, Fig S4c is not labeled properly.
3. Intra-DLS infusions of rimonabant also reduce ketamine-induced locomotor activity, but these effects are only observed at sessions 5-7, not earlier? Why are the effects delayed? This is not the case for the effects on self-administration.
4. What was the ketamine dosing for the morphological studies of SPN neurons in Figure 3? Since this study uses multiple ketamine dosing schedules the exact paradigm should be shown for each figure and stated in the text.
5. In the studies of MAGL immunohistochemistry (Figure 4d,e), why is the bar graph labeled as relative PRDM5 levels? It is troubling that this type of error in labeling and/or data occurs throughout the results section.
6. For the studies of viral expression of MAGL \pm ketamine \pm 2-AG (Figure 5 g,h), what is the effect of 2-AG alone on locomotor activity?
7. The rationale for targeting PRDM5 is stated in the Discussion, but this should be given earlier. Are the effects for ketamine regulation of PRDM5 specific or are other transcription factors also regulated?
8. For the primary cell culture studies, the concentration of ketamine, 100 μ M that produces a significant effect on levels of MAGL and PRDM5 is not in the range that would be reached after systemic administration of ketamine, raising concerns about the relevance of these cell culture results and making interpretation of the findings questionable.
9. Since it is difficult to demonstrate a direct effect of ketamine on SPN neurons that would lead to increased 2-AG levels via downregulation of MAGL and up-regulation of PRDM5, what other possible mechanisms could account for these effects?

Response letter:

Reviewer #1 (Remarks to the Author):

In this study, Xu and collaborators provide strong data regarding the impact of ketamine on the endocannabinoid system in a mouse model. Using a combination of lipidomic, cell morphological, pharmacological, in vivo gene manipulation and behavioral tools, the authors demonstrate that endocannabinoids, and particularly 2-AG, play a critical role in the psychostimulant and reinforcing effects of ketamine, especially in the caudate-putamen (CPu). The data also show that MAGL, the hydrolytic enzyme of 2-AG, is regulated by negative transcriptional factor PRDM5, revealing a potential mechanism for ketamine-seeking behavior. Overall, the study provides important information about the neurobiological effects of ketamine, a drug used in the clinic for different applications but also abused in a non-medical environment. In addition, the data highlight the endocannabinoid system as a potential novel therapeutic target for ketamine addiction that supports published data that has demonstrated a significant effect of ketamine and glutamatergic regulation on the endocannabinoid system. The manuscript is presented well and the experimental approaches as well as the results are solid. The manuscript could be improved by addressing the following suggestions:

1. Overall, the data is strong but the literature is not well cited since it has already been shown that ketamine alters the endocannabinoid system and strongly established of the glutamatergic regulation (both metabotropic and NMDA) of endocannabinoid levels.

Reply: As the reviewer suggested, we added literatures regarding the glutamatergic regulation of endocannabinoid system in the Discussion (Page 21-22, Line 593~603). Also, these literatures are listed in the Reference (references 56~59) in revised manuscript (abbreviated as revised MS).

2. The Abstract states that CPu 2-AG production may be a potential novel therapeutic target for ketamine addiction. How can CPu PRMD5 truly be a rational therapeutic target?

It would be best to rephrase this sentence and other similar lines in the paper.

Reply: We deleted this statement from the Abstract, and we also revised other similar statement in the revised manuscript. In general, it is described as follows: Manipulation of 2-AG metabolic enzymes may serve as potential therapeutic targets for the treatment of ketamine addiction (Page 22, Line 612~614, in revised MS).

3. The last paragraph of the Introduction mentions genes that should be described in a bit more detail for the general audience at the start: please define that *Mgll* is the gene encoding for MAGL and the known role of PRDM5.

Reply: We thank reviewer for these helpful suggestions and added a few description of PRDM5 and *Mgll* in the Introduction (Page 4, Line 89~97).

4. PRDM5 was chosen to be one focus of the study (experiments in Figs. 6-8) but there is no rationalization as to why this particular transcription factor was studied considering all the various transcriptional regulatory mechanisms that are expected to play a role in regulating *Mgll*. Please explain the rationale in the context of specific functions.

Reply: We added the information regarding the the rationale why we chosed PRDM5 for exploring the transcriptional regulatory mechanism of *mgll* gene in the Introduction (Page 4, Line 94~97), as well as in the Results (Page 14, Line 395~396) in revised manuscript.

5. No rationale is give as to why the 15 mg/kg ketamine dose was chosen for the experiments. It would be important to justify, e.g. that it is the equivalent of ketamine commonly used recreationally. Also, what was the dosing schedule in the morphological experiments presented in Fig. 3?

Reply: We chosed the dose of 15 mg/kg for the experiments based on the following reasons:

(1) Ketamine abusers inject ketamine usually at a dose of 100~200 mg/person [1]. Assuming 60 kg per an adult, the dose of ketamine abusers is 1.67~3.33 mg/kg. Considering that biological equivalent dose coefficient (mice/human=9.1), 15~30 mg/kg in mice is almost equivalent to recreational use in human.

(2) In animal research, ketamine increases locomotor activity at the dose of 4~16 mg/kg [2]. Another animal research also showed that 30 mg/kg ketamine induces locomotion sensitization in mice [3].

(3) In fact, we had detected the effect of different doses of ketamine on behavior at the beginning of our research. We found that both 15 and 30 mg/kg ketamine obviously increased locomotor activity in mice (Supplementary Fig S3B). Thus, we chose 15 mg/kg ketamine for the experiments.

As for the dosing schedule in the morphological experiments, we used 15 mg/kg ketamine in Fig. 3 and added dose of ketamine in the revised manuscript (Page 9, Line 247~248) and Figure legend (Page 30~31, Line 867~868, Line 887).

6. Table 1 shows a lot of negative changes especially in the CPU. Please provide a rationale as to why the focus of the rest of the paper is on increased endocannabinoid levels.

Reply: We thank review for this suggestion. There were a lot of lipids modified by ketamine. Clearly, it is impossible for us to investigate their potential roles in mediating ketamine effect. In light of the critical roles of eCBs in drug reward as well as the significant 2-AG alterations, we focused on their functions in mediating ketamine effect. We added some discussion regarding the modified lipids in the Discussion (Page 18~19, Line 504~515) in the revised MS.

7. It is difficult to read the names of the different lipids presented in Fig. 1c. For simplicity, this could be moved to the Supplement. The results shown in Fig 1i and 1k seem to be switched in how they are described in the text on p7, since it refers to Fig. 1k for a single ketamine dose and Fig. 1i for repeated administration when the figure shows the opposite.

Reply: Thank for this suggestion. We moved Fig 1a to the Supplement (Supplementary Fig. S1D in the revised MS). We also rephrased the corresponding text regarding Fig. 1k and Fig. 1i (Fig 1i and Fig 1g in the revised figures matches the original Fig. 1k and Fig. 1i, respectively) in the revised manuscript (Page7, Line177~181).

8. The focus on 2-AG instead of AEA to ketamine addiction is based on the significant 2-AG alterations observed in the hyperlocomotion and self-administration models. However, in Fig. 1f, the lack of significant effect of ketamine to increase AEA is actually due to very large variability in the saline animals that is not evident in other brain areas. It raises questions about technical issues especially since there was no such variability in saline animals in Fig. 1h where significant AEA elevation in the CPu is induced by ketamine. Given the link of glutamatergic transmission with AEA, one also wonders whether levels of AEA-related enzymes such as FAAH are altered which could provide fundamental knowledge about the endocannabinoid pathway(s) relevant to ketamine.

Reply: We thank the reviewer for the constructive suggestion. We re-analyzed the original data of AEA by deleting the minimum and the max values in the saline group. The results showed that ketamine did not statistically increase AEA level (Fig. 1d in the revised figure). We then added a new immunoblotting test to detect the protein expressions of AEA-related enzymes, such as FAAH and NAPE-PLD. We found no significant alterations in the expression of these two enzymes. We supplemented these new data in the Fig S9B in the Supplementary information.

9. In Fig.2b, rimonabant seems to have an effect on hyperactivity starting at the fifth session only. Is there any explanation for this delay?

Reply: We thank the reviewer for this constructive question. In hyperlocomotion paradigm, the mice were passively received 15 mg/kg ketamine daily. We guess that 1 $\mu\text{M}/\mu\text{L}/\text{injection}$ rimonabant may not be high enough to reverse ketamine effects in hyperlocomotion paradigm. To test this idea, we supplemented a new study. We increased the dose of intra-DLS rimonabant to 0.6 $\mu\text{g}/\mu\text{L}/\text{injection}$ and tested its effect on ketamine-induced hyperactivity in mice. We observed that rimonabant (0.6 $\mu\text{g}/\mu\text{L}/\text{injection}$) clearly inhibited ketamine-induced hyperactivity starting at the first session. We supplemented this new data in Fig 2b in the revised manuscript.

10. The causal data regarding Mgl1 knockdown is very strong. Is there also evidence on

the individual level that ketamine disrupts the normal relationship between the behavior and molecular regulator for the genes and proteins that were studied in the animals in Figs. 1e-1h and 2h-2k?

Reply: So far, we have not seen direct epidemiological evidence between MAGL/ PRDM5 and human abuse. A study showed that alcoholic subjects present the lower MAGL activity [4] and few studies showed that eCBs may be associated with post-traumatic stress disorder (PTSD) or decision making [5, 6]. In addition, inhibition of MAGL increases cue-induced reinstatement of nicotine-seeking behavior in mice [7]. We will pay attention to this link in the future.

11. Fig.4j shows that MAGL expression is still downregulated at 72h, whereas the levels of 2-AG are back to baseline at the same timepoint (Fig.1k). In this case, restored 2-AG level would not be consistent with MAGL downregulation. Is there any compensatory mechanism arising? While not extremely important for the overall conclusions of the study, the levels of enzymes involved in 2-AG biosynthesis could be measured at the same timepoint. Please at least provide a possible explanation.

Reply: Thanks for this constructive comment. As suggested by the reviewer, we performed a new experiment to explore whether there would be a potential compensatory mechanism for 2-AG production. By westernblotting we detected the expression of CPU DAGL α , an important enzyme for 2-AG biosynthesis, at 0, 6, 24, 48 and 72h after acute or repeated ketamine administration, respectively. The results showed that DAGL α expression was reduced 72h after the last injection of repeated ketamine, but without obvious DAGL α expression change at other time points, suggesting that a compensatory mechanism for 2-AG production may arise. We added this new result in the revised manuscript (Page11~12, Line313~318; Supplementay Figure S11A, B).

12. There is a confusion around the symbols for the different treatments in Fig.S4c. Please use the correct shapes and colors for the definitions.

Reply: We improved the symbols for the different treatments in Fig.S4c.

13. Lines 356-364 (information on Fig. 5d-l) show experimental validation and might be more logical to describe before the other tests that use these approaches to address behavioral effects.

Reply: The levels of eCBs were detected in two ketamine addiction models, and it would be more reasonable to describe the models, followed by eCBs detection. If the model fails, it may not be necessary to detect eCBs.

14. It is stated that infusion into the dorsolateral striatum with 2-AG shifted the ketamine dose response curve. However, the effect is rather small and only observed at one dose. This and other results in the figures highlight the fact that no F-values or other outcome from statistical models are provided in the Result section.

Reply: We thank the reviewer for the suggestion. Fig. 5p showed that 2-AG significantly sensitized response to 0.03 mg/kg/infusion ketamine as well as an increasing response to 0.1 mg/kg/infusion ketamine (Fig. 5p). There is a similar research focusing on cocaine, which also showed a similar and small shift of cocaine dose-response curve (Fig. 2e, 4f, reference [8]).

Detailed statistical values, including F-values, p value and statistical models, are included in all figure legends and in the source data.

15. In Fig.6j, are these all shRNA instead of si?

Reply: They are shRNA instead of siRNA (Fig. 6j).

16. In Fig. 6n and o, adding “2-AG” in the title of the y axis would make it easier to understand what was done.

Reply: We added 2-AG in the title of y axis in Fig. 6n and 6o. In addition, we added 2-AG or AEA in the title of y axis in all graphs (Fig. 1c-j, Fig. 5d, l, Fig. 6n, o, Fig. 7c; Supplementary, Fig. S12B, C, F, Fig. S15, in the revised figures).

17. Fig. 8 shows many epigenetic mechanisms but none of them were examined in the study, so it is excessive and not very informative. To make the summary more meaningful,

perhaps show two simpler versions: the normal regulation vs. ketamine effects next to each-other. It would be more visually logical to show pre-synapse on the top and post-synapse on the bottom. Moreover, there is not sufficient published evidence for 2-AG binding to AMT, so that could be deleted.

Reply: Thank reviewer for this helpful suggestion. We modified and separated the mechanistic graph in the revised manuscript. We also deleted AMT (Fig. 8a and 8b).

18. It is stated in the Discussion, line 450, that ketamine increases MAGL, when it actually reduced it.

Reply: It was found to be a typographical error. We corrected it (Page 18, Line 486).

19. Lines 488-493: it is an over-speculation that there is a dose-dependent effect of 2-AG on a specific cell-type specific as there is not sufficient data showing this and even the potential “shift” of CB1R activation between glutamatergic and GABAergic terminals.

Reply: We agree with reviewer's comment and deleted this part from the Discussion in the revised manuscript .

20. It would be helpful to provide information about the drug doses used in the figure legends.

Reply: We added the information of drug dosages in the figure legends involved.

Response References:

1. AL, W., et al., *Ketamine abusers presenting to the emergency department: a case series*. The Journal of emergency medicine, 2000. **18**(4): p. 447-51.
2. G, I., et al., *Dose-response characteristics of ketamine effect on locomotion, cognitive function and central neuronal activity*. Brain research bulletin, 2006. **69**(3): p. 338-45.
3. JP, G., et al., *Profiling of behavioral effects evoked by ketamine and the role of 5HT and D receptors in ketamine-induced locomotor sensitization in mice*. Progress in neuro-psychopharmacology & biological psychiatry, 2020. **97**: p. 109775.
4. Erdozain, A.M., et al., *The endocannabinoid system is altered in the post-mortem prefrontal cortex of alcoholic subjects*. Addict Biol, 2015. **20**(4): p. 773-83.
5. Crombie, K.M., et al., *Loss of exercise- and stress-induced increases in circulating 2-arachidonoylglycerol concentrations in adults with chronic PTSD*. Biol Psychol, 2019. **145**: p.

- 1-7.
6. Hill, M.N., et al., *Reductions in circulating endocannabinoid levels in individuals with post-traumatic stress disorder following exposure to the World Trade Center attacks*. *Psychoneuroendocrinology*, 2013. **38**(12): p. 2952-61.
 7. Trigo, J.M. and B. Le Foll, *Inhibition of monoacylglycerol lipase (MAGL) enhances cue-induced reinstatement of nicotine-seeking behavior in mice*. *Psychopharmacology (Berl.)*, 2016. **233**(10): p. 1815-22.
 8. Anderson, E.M., et al., *Overexpression of the Histone Dimethyltransferase G9a in Nucleus Accumbens Shell Increases Cocaine Self-Administration, Stress-Induced Reinstatement, and Anxiety*. *J Neurosci*, 2018. **38**(4): p. 803-813.

Reviewer #2 (Remarks to the Author):

The authors examined roles of endocannabinoids in the psychoactive effects of ketamine using a combination of lipidomic, biochemical, molecular, morphological and behavioral approaches to assess endocannabinoid levels and their roles in ketamine locomotor activation and self-administration. The self-administration approach is especially powerful as it indicates mechanisms that might be involved in abuse of the drug. The authors also explored the role of factors that control expression of endocannabinoid metabolizing enzymes. Effects of chronic versus acute exposure were also elucidated. Overall, the manuscript presents new and intriguing findings linking endocannabinoids to ketamine actions. However, there are significant questions about the link between ketamine and changes in endocannabinoid function, as well as concerns about some of the experimental approaches, reagents and interpretations. In addition, the manuscript needs significant revision to clarify several points.

Major Comments

1. It is not clear if the ketamine interactions with the endocannabinoid system are secondary to initial actions at the NMDA receptor or another molecular target. Given the current controversies about the locus of initial ketamine and ketamine metabolite effects on the brain it would be good to address this question.

Reply: We thank reviewer for this valuable suggestion. As reviewer pointed out, there are controversies regarding the locus of initial ketamine and ketamine metabolite effects on the brain. To try to address reviewer's question, we applied MK801 to investigate whether it would exhibit the similar effect with ketamine, as both ketamine and MK801 are NMDAR antagonist. We added several new studies as follows:

(1) The effect of MK801 on ECS and behavior was investigated in mice. The result showed that MK801 enhanced locomotor activity of mice. Similar with ketamine' effect, MK801 also decreased MAGL expression and increased 2-AG level in the DLS (Supplementary, Fig. S12A-D).

(2) The effect of MK801 on MAGL expression and 2-AG level in cultured SPNs was

investigated. The result showed that MK801 decreased MAGL expression, and increased PRDM5 expression as well as 2-AG level. These effects are well in line with ketamine's (Supplementary, Fig. S12E, F).

Above results show that MK801 exhibit similar effect on MAGL/PRDM5 expression and 2-AG level. Thinking that both ketamine and MK801 are NMDAR antagonist, we infer that ketamine action may be mediated, at least in part, by NMDAR. We added above new data in the Results (Page 12, Line 319~326; Page 17, Line 474~479) in the revised manuscript.

2. The rationale for examining CB1 coimmunoprecipitation with $G_{\alpha/i}$ is not clear. In general, activation of GPCRs should decrease, not increase, interaction with G_{α} subunits, and this occurs on a very rapid timescale that may not be captured by a technique with temporal resolution like Co-IP. Why rimonabant would block this increased Co-IP is also unclear, as it should, if anything, promote increased affinity for $G_{\alpha/i}$ by itself. Overall, it seems unclear that the authors know how GPCRs function.

Reply: We agree with reviewer's opinion that CB1 coimmunoprecipitation with $G_{\alpha/i}$ may not be a appropriate technique because of so rapid interaction of these two proteins. The reason why we had performed this Co-IP study was to show a direct effect of ketamine on CB1R function. Thinking of the technical issue, to avoid confusion, we removed this part (CB1 coimmunoprecipitation with $G_{\alpha/i}$, both in the text and figures involved) from the revised manuscript.

CB1R is one kind of GPCRs coupled with $G_{\alpha/i}$. It has been known that CB1R activation inhibits adenylyl cyclase activity and downregulate cAMP level. Thus, cAMP reflexes the status of CB1R activation and is used as an indicator of CB1R function [1]. To explore the effect of ketamine on CB1R function, we added a new study to quantitatively detect cAMP content in the DLS after ketamine treatment, and rimonabant, a selective CB1R inverse agonist, was used as control. We found that ketamine significantly reduced cAMP level whereas rimonabant elevated cAMP level in the DLS under ketamine hyperlocomotion paradigm. The result showed that ketamine activates CB1R function. We added this new data in the revised manuscript (Fig. 2c).

3. In the initial lipidomics screen the data need to be corrected for multiple comparisons and/or false discovery rate needs to be calculated.

Reply: We analyzed the lipidomics data by Progenesis QI software (Nonlinear Dynamics, Waters Corp., USA), a professional and widely used lipidomic analysis software, in which statistical analysis function has been integrated. Based on study design, such as group number, sample number/per group and time points, the statistical analysis is automatically performed. The details of algorithm is provided online (<http://www.nonlinear.com/progenesis/qi/how-it-works/>).

4. The locomotor effects of some treatments appear to be only in the later test sessions (e.g. figures 2b and 6k), while in other experiments the effects are there from the beginning of treatment. Can the authors explain this apparent discrepancy?

Reply: We thank the reviewer for the suggestion. We infer that this discrepancy may be largely due to the dosage of rimonabant and the different administration route. To address this question, we supplemented a new experiment of mice hyperlocomotion by using a high dose of rimonabant (0.6 μg per DLS). The result showed that 0.6 $\mu\text{g}/\text{DLS}$ rimonabant effectively prevent ketamine-induced hyperlocomotion at the first session. We added this new result in the revised manuscript (Fig. 2b).

In addition, in self-administration paradigm, mice get drug by active nosepoke. The dose of 1 μM rimonabant was able to block the reward induced by occasionally ketamine administration (0.5 mg/kg/infusion). But in hyperlocomotion paradigm, the animals passively received a fixed high dose of 15 mg/kg ketamine, We speculate that 1 μM rimonabant may not be sufficient enough to reverse hyperlocomotor effect of ketamine at a high dose paradigm (15 mg/kg/day, continuous injection for 7days).

5. Sample sizes appear to be very small in some experiments (e.g. figure 1h-i). Did the authors perform a power analysis to determine what would be needed to support or reject the null hypothesis?

Reply: Considering the small sample sizes may lower the detection power, we re-performed these experiments with larger sample size (9 mice per group). The results

are in consistent with the previous results: 2-AG level decreases within 48 hours after the last injection of related ketamine exposure (Fig. 1 g-j in the revised figures matches the original Fig. 1i-l, respectively).

In addition, the sample sizes in our original experiments are in line with literatures (6-10 animals per group) [1-3]

6) Figures 4d and 7f, why is MAGL so strongly expressed in somata and dendrites? In most brain regions this enzyme is mainly in presynaptic terminals. How reliable is the MAGL antibody?

Reply: MAGL antibody we used was from Cayman (#100035) and validated by the producer. Previous studies also show that MAGL is strongly expressed in somata and dendrites in cortical neurons [4]. In addition, MAGL immunostaining for glioma cells with other two MAGL antibodies (Millipore, ABN1000; abcam,#246902) also shows MAGL expression in the somata and dendrites [5].

7) In the pERK immunoblots only the lower molecular weight form is increased. It is not clear what this should happen, but it might be due to insufficient substrate in the assay.

Reply: We thank reviewer for this constructive comment. We re-performed the immunoblotting and found that both low and high molecular weight forms of pERK are increased. We added this new results in the revised manuscript (Fig. 2d) and removed the original one.

8) What was the source of the MAGL used in the enzyme activity assay?

Reply: Human recombinant MAGL was contained in the assay kit (Monoacylglycerol Lipase Inhibitor Screening Assay Kit, Cayman, #705192). The kit details are listed at the Producer's website (<https://www.caymanchem.com/pdfs/705192.pdf>).

9) Was expression of virally-delivered constructs specific for neurons?

Reply: We used CAG promotor which is ubiquitous and strong to construct AAV overexpressing MAGL. Although CAG promoter is not specific for neurons, our data

showed that neurons are predominantly responsible for ketamine-evoked ECS alterations: (1) Immunostaining of mice CPu showed that MAGL expression was mainly reduced in neurons (Fig.4d,e). (2) In vitro results in primary cultured striatal neuron were consistent with the results *in vivo*, suggesting that neurons may play predominant role in mediating ketamine's effects (Fig. 7a-j).

10) Figure 6c, is the red labeling supposed to be visible in the MERGE images? If so, it is not.

Reply: The red color is labeled for PRDM5 and the green labeled for NeuN. Both PRDM5 and NeuN are located in the nuclear. Thus, the merged color exhibits yellow in the merged image (Fig. 6c, arrows indicate).

11) Why was a CMV promoter used for the shRNA construct? In general, this promoter is not that strong in neurons in situ, and may also drive expression in glia.

Reply: Sorry for the unclear description about the promoter of shRNA construct. In fact, CMV promoter was used to express eGFP in control AAV, and hU6 promoter used for AAV-shPRDM5 in our study. We corrected this mistake (Supplementary Fig. S17; Page 5, Line143~144) in the revised manuscript. In addition, in recent years, more and more studies have applied hU6 promoter for gene interference, as hU6 promoter is bound by RNA pol III and is the most optimal promoter to produce shRNAs [6].

12) The authors need to provide information on the specificity of all the antibodies used in the study. Preferably, specificity should be shown using knockout mouse tissue. Also, were protease inhibitors used in the tissue preparation and blotting procedures (they should be). What were the secondary antibodies for the striatal SPN culture experiments?

Reply: All the antibodies used in this study were purchased from commercial companies. The secondary antibodies used in striatal SPN culture experiments were the same as those used in animal experiments. We provided the detailed information of all the antibodies, including the secondary antibodies for striatal SPN culture, in the Supplementary Table 5 in the revised manuscript.

To prepare tissues for immunoblotting, we used protein extraction kit (#K269, Biovision, Milpitas, CA, USA), which contains protease inhibitor cocktail and DTT in the kit.

13) In the ChiP-qPCR assay, how much of the material from the punches comes from SPNs, and could there be a signal-to-noise ratio problem due to the use of this approach?

Reply: To answer this question, we conducted flowcytometry and Immunohistochemistry to figure out SPNs ratio in the striatum. We collected CPu tissue from 9 mice and separated them into single cell. We then performed immunostainign for DARPP32, NeuN and GFAP, respectively, and we also counted the positive cell number by flowcytometry. The results showed that DARPP32-positive cells possessed ~31%, NeuN-positive cells ~32.6%, and GFAP positive cells ~23.1% (Supplementary, Fig. S22A). Immunohistochemistry detection also showed that DARPP32-positive cells roughly possessed 38% of the total cells (Supplementary, Fig. S22B, C). Therefore, SPNs (DARPP32-positive) are the predominant cell type in CPu tissue. In addition, by using an immunostaing method, a previous study showed that DARPP32-positive cells accounted for ~68% in dorsal striatum [7]. We infer that such difference may be due to the different methods used. In spite that there is difference regarding the ratio of DARPP32-positive cells, our data together the literature indicate that SPNs are the main cell type in the CPu.

In addition, we adopted two measures to technically control the technical quality. Firstly, we accurately weighed 30 ± 0.1 mg tissue (collected from 3 mice) per sample to perform ChIP. Secondly, we used IgG as negative control to reflect signal-to-noise. From Fig. S16b, we can see the percentage of anti-IgG lowered than 0.05%, reflecting a good signal-to-noise control (Page3, <https://media.cst-c.com.cn/pdf/9005.pdf>).

Minor Comments:

a. Page 3, the statement that "...the precise mechanisms underlying the psychopharmacological effect of ketamine is largely unknown." is an overstatement. Several primary targets for ketamine action are known, the NMDAR being the most

prominent one. Downstream effects may not be as clear, but at least we know a great deal about initial actions in the brain.

Reply: We modified this statement in the text accordingly (Page 3, Line 69~70) in the revised manuscript as follows: “ Despite these advances, the downstream effects of ketamine are not fully elucidated.”

b. The last paragraph of the introduction may confuse readers. A brief description of PRDM5 and the rationale for examining this should be provided earlier in this section to provide the context for the last sentence.

Reply: We thank the reviewer for the helpful suggestions and added a few description of PRDM5 in the Intruduction (Page 4, Line 89~97, revised MS).

c. Several potentially interesting changes in lipid expression, including some decreases, are noted in table 1 (provided they survive proper analysis). Clearly, these changes cannot be examined in any detail in this study, but they should at least be noted in the discussion.

Reply: We added discussions regarding the changes of other lipids in the Discussion (Page18~19, Line 504~515, revised MS).

d. For the locomotor activity analysis were animals always examined in the same environment, or was a novel environment used each time?

Reply: The mice were examined in the same enviroment in the locomotor activity analysis.

e. What tissue fixation, if any, was used in the Golgi-Cox procedure?

Reply: We performed DLS Golgi-Cox staining by using the commercial kit (FD Neuro Technologies, USA), according to the manufacturer's instructions (http://www.fdneurotech.com/docs/1333571253.web_pk401-401a-04042012.pdf). We mixed solution A and solution B (1:1) for fixation. Solution A should be 5% potassium dichromate and solution B be 5% mercuric chloride.

f. Why is PRDM5 data presented in figure 4e when this protein has not yet been discussed? Is this graph mislabeled?

Reply: It was found to be a typographical error, and the correct one is MAGL. We corrected this error in Fig. 4e.

g. Figures 6 and 7, the y-axis label "Relative Level to Internal Standard" is uninformative. Please indicate what was actually measured (e.g. 2-AG in figure 6n).

Reply: We added 2-AG or AEA in the title of y axis in all graphes regarding eCB detection (Fig. 1c-j, Fig. 5d, l, Fig. 6n, o, Fig. 7c; Supplementary, Fig. S12B, C, F, Fig. S15) in the revised manuscript.

h. Figure 7b, should the units be uM rather than umol?

Reply: It was a typographical error, and the units should be μM . Thinking that ketamine (100 μM) concentration may be too high and cause false-positive results, we added a new experiment of primary cultured SPNs *in vitro* with low concentration of ketamine (1.5 $\mu\text{g/mL}$, equal to 5.5 μM) treatment, followed by measurement of MAGL and PRDM5 expressions by immunoblotting and 2-AG detection. We found that ketamine (1.5 $\mu\text{g/mL}$) also significantly decreased MAGL expression and increased PRDM5 expression as well as 2-AG level in the cultured SPNs. The result was consistent with that of high concentration of ketamine (100 μM). We adopted the updating data (Fig. 7a-c) to replace the original one in the revised manuscript.

i. Figure S17b, at what DIV were these neurons examined. The DAPI staining seems to indicate pretty sparse neuronal expression.

Reply: The neurons were examined at DIV7. The sparse neuronal expression was due to the low-density culture of neurons as well as the high magnification of image in this study. We replaced the image with new low magnification image (Supplementary, Fig. 20B) in the revised manuscript.

j. Page 7, end of first paragraph, at this point the data only show a correlation with

increased endocannabinoid expression, so this last sentence should be reworded.

Reply: According to reviewer's suggestion, we modified the text as follows: "hinting the potential effect of these two lipids on ketamine behavioral effect" (Page 7, line 171~172, revised MS).

k. Rimonabant should be referred to as an inverse agonist.

Reply: We thanks the reviewer and used the accurate description of Rimonabant, "an inverse agonist", in the revised manuscript (Page 8, Line 196; Page 20, Line 557).

l. Page 8, first paragraph, make it clear that the PKA and CREB assays were performed on tissue from animals that self-administered ketamine, thus the effect may be due to the combination of the drug and the learning itself.

Reply: According to reviewer's suggestion, we modified the text as follows: "As the effects of ketamine on PKA and CREB phosphorylation differed in the hyperlocomotion and self-administration paradigm, we speculated that it may be due to the combination of psychopharmacological effect and learning induced by ketamine" (Page 9, line 239~242, in revised MS)

m. Page 9 and 10, the authors should first describe the ketamine-induced changes in dendritic arborization before describing the rimonabant block.

Reply: We modified the text and first described ketamine-induced changes in dendritic arborization before describing the rimonabant blockade (Page10, line 260~261, in revised MS).

n. While the term "clastic" is technically correct for description of these enzymes, it is a rather old-fashioned term. Perhaps catabolic, degradative, or another term would be better.

Reply: We replaced "clastic" with "catabolic" (Page 10, Line 278, in revised MS).

o. Page 16 last paragraph, make it clear that the nuclear labeling was used for PDRM5

localization while dendritic localization with MAP2 was used to examine MAGL subcellular location.

Reply: We initially used MAP2 to outline neurons (dendrite). But the fluorescence of MAP2 (1:10000 dilution) was too intense and interfered the evaluation of MAGL. Considering that this data is dispensable, we deleted this part from the manuscript.

p. Supplement, line 56, I assume the authors mean to say ketamine and not cocaine.

Reply: Yes, we mean ketamine, and it was a typographical error. We corrected it (Supplementary, Page 1, Line 23, in revised MS).

q. Supplement, line 162, the authors indicate that only 1 week was given between AAV injections and behavioral testing onset. Is this time period sufficient for viral infection and expression?

Reply: Sorry for this unclear description. In fact, the mice recovered 3 weeks between AAV injections and behavioral testing onset, as you can see from the Fig. 5e,g,i. We revised the text in the Method (Supplementary, Page 4, Line 110~113).

r. There are several grammar and usage errors in the manuscript, especially in the supplemental material. Additional editing is required. For example, there is no “Turkey’s” post hoc test.

Reply: Thank for this suggestion. We carefully checked the revised manuscript, especially Supplemental Material, to minimize typographical and grammatical errors. In addition, we checked the statistical methods of our data, and found that one-way ANOVA had been performed by SPSS 19.0 software and two-way ANOVA been performed by GraphPad Prism 7. To unify the statistical method in this study, we re-analyzed some data with GraphPad Prism 7, and the statistical results are almost same with the previous results.

Response references:

1. Hebert-Chatelain, E., et al., *A cannabinoid link between mitochondria and memory*. *Nature*, 2016. **539**(7630): p. 555-559.

2. Anderson, E.M., et al., *Overexpression of the Histone Dimethyltransferase G9a in Nucleus Accumbens Shell Increases Cocaine Self-Administration, Stress-Induced Reinstatement, and Anxiety*. J Neurosci, 2018. **38**(4): p. 803-813.
3. Strong, C.E., et al., *Locomotor sensitization to intermittent ketamine administration is associated with nucleus accumbens plasticity in male and female rats*. Neuropharmacology, 2017. **121**: p. 195-203.
4. Dinh, T.P., et al., *Brain monoglyceride lipase participating in endocannabinoid inactivation*. Proc Natl Acad Sci U S A, 2002. **99**(16): p. 10819-24.
5. Liu, Q., et al., *Effects of co-administration of ketamine and ethanol on the dopamine system via the cortex-striatum circuitry*. Life Sci, 2017. **179**: p. 1-8.
6. Henriksen, J.R., et al., *Comparison of RNAi efficiency mediated by tetracycline-responsive H1 and U6 promoter variants in mammalian cell lines*. Nucleic Acids Res, 2007. **35**(9): p. e67.
7. Matamales, M., et al., *Striatal medium-sized spiny neurons: identification by nuclear staining and study of neuronal subpopulations in BAC transgenic mice*. PLoS One, 2009. **4**(3): p. e4770.

Reviewer #3 (Remarks to the Author):

This study examines the role of 2-AG in the locomotor activating and self-administration effects of ketamine. The results demonstrate that repeated ketamine administration or self-administration increases levels of 2-AG in the dorsal lateral striatum, and that this effect is mediated by decreased expression of the metabolizing enzyme MAGL. Further, the results show that the decrease in MAGL is mediated by increased levels of the transcription factor PRDM5, a negative regulator of MAGL gene expression. The studies include evidence that overexpression of MAGL or knockdown of PRDM5 via viral expression of shRNA block the effects of ketamine on locomotor activity or self-administration, although with differential effects depending on which is targeted. In general, these studies are well designed and the results provide evidence that regulation of 2-AG, MAGL, and PRDM5 are involved in the actions of ketamine, although there are several points to address.

1. The results section states that levels of 2-AG were significantly increased at 6, 24, and 48 h after the last ketamine injection, but this is not consistent with what is shown in Fig. 1i and Fig 1k? This should be addressed.

Reply: We re-edited the figures. Fig. 1g and Fig. 1i in the revised MS replaced original Fig. 1i and Fig. 1k, respectively. Fig. 1g showed the change of 2-AG after single ketamine exposure, and Fig. 1i showed the change of 2-AG after repeated ketamine exposure. To make it clear, we added a few description regarding these two figures in the revised manuscript (Fig. 1g~j; Page7, Line 177~181).

2. The rimonabant studies are confusing as higher systemic doses (1-3 mg/kg, ip) decrease but lower doses (0.1 and 0.3 mg/kg) increase locomotor activity. This is in contrast to what is stated in the text of the results. In addition, the dose of 0.6 mg/kg that blocks the locomotor effects of ketamine also decreases activity. Also, Fig S4c is not labeled properly.

Reply: In Fig. S4A, we tested the effect of rimonabant (1, 2, 3 mg/kg) on baseline mice

locomotion. We found that 3 mg/kg rimonabant inhibited mice locomotion since Day 4, and rimonabant (1、 2 mg/kg) inhibited locomotion since Day 6.

Thinking that the dose of rimonabant in Fig.S4A test was high, we re-tested the effect of low doses of rimonabant (0.1、 0.3、 0.6 mg/kg) on mice baseline and ketamine-induced locomotion (Fig. S4B). The mice were treated with rimonabant for 10 days continuously, and together treated with ketamine treatment in the last 4 days. In the first 6 days, rimonabant (0.1、 0.3、 0.6 mg/kg) showed no effect on mice baseline locomotion. Since Day 7, rimonabant (0.1、 0.3 mg/kg) showed no effect on ketamine effect, but 0.6 mg/kg rimonabant significantly inhibited ketamine's effect. This result showed that low dose of rimonabant (0.6 mg/kg) effectively attenuate ketamine-evoked hyperlocomotion without affecting the baseline locomotor activity.

We rephrased this part (Supplementary, Page 16, Line 424~433) and re-labeled Fig. S4C properly.

3. Intra-DLS infusions of rimonabant also reduce ketamine-induced locomotor activity, but these effects are only observed at sessions 5-7, not earlier? Why are the effects delayed? This is not the case for the effects on self-administration.

Reply: We infer that this discrepancy may be largely due to the dosage and different administration route of rimonabant. In self-administration paradigm, mice get drug by active nosepoke, and the dose of 1 μ M rimonabant may be sufficient to block the reward effect induced by occasionally ketamine administration (0.5 mg/kg/infusion). But in hyperlocomotion paradigm, the mice passively received a fixed high dose of 15 mg/kg ketamine. We thus speculate that 1 μ M rimonabant may not be sufficient enough to reverse hyperlocomotor effect of ketamine at a high dose (15 mg/kg/day, continuous injection for 7days). We also supplemented a new experiment studying the effect of a high dose of rimonabant (0.6 μ g per DLS) on ketamine-induced hyperlocomotion. The result showed that 0.6 μ g/DLS rimonabant effectively prevent ketamine-induced hyperlocomotion at the first session. We added this new result in the Fig. 2b in the revised manuscript.

4. What was the ketamine dosing for the morphological studies of SPN neurons in Figure 3? Since this study uses multiple ketamine dosing schedules the exact paradigm should be shown for each figure and stated in the text.

Reply: We supplemented the exact paradigm in the figures (Fig.3) and modified the text (Page 9, Line 247~248) and figure legend (Page 30~31, Line 867~868, Line 887) in the revised manuscript accordingly.

5. In the studies of MAGL immunohistochemistry (Figure 4d, e), why is the bar graph labeled as relative PRDM5 levels? It is troubling that this type of error in labeling and/or data occurs throughout the results section.

Reply: It was found to be a typographical error. We carefully checked all the figures and text, and corrected the errors. We make sure the correct labeling in all figures.

6. For the studies of viral expression of MAGL ± ketamine ± 2-AG (Figure 5 g,h), what is the effect of 2-AG alone on locomotor activity?

Reply: Thank for this constructive suggestion. To address this question, we added a new study to investigate the effect of 2-AG along on locomotor activity in mice. The result showed that 2-AG (10 μM) along showed no effect on locomotor activity. We added this new data in the revised manuscript (Supplementary Fig. S14)

7. The rationale for targeting PRDM5 is stated in the Discussion, but this should be given earlier. Are the effects for ketamine regulation of PRDM5 specific or are other transcription factors also regulated?

Reply: We thank the reviewer for this instructive suggestion and added a few description regarding PRDM5 in the Introduction (Page 4, Line 89~97). So far, We have no evidence about other transcription factors that may be involved in the regulation of gene *Mgll*. Our results indicate that PRDM5 is one of the important transcriptional factors for *Mgll*. We will look at other potential transcription factors in the future.

8. For the primary cell culture studies, the concentration of ketamine, 100 μM that

produces a significant effect on levels of MAGL and PRDM5 is not in the range that would be reached after systemic administration of ketamine, raising concerns about the relevance of these cell culture results and making interpretation of the findings questionable.

Reply: We thank the reviewer for the instructive suggestion. Thinking that ketamine (100 μM) concentration may be too high and cause false-positive results, we added a new experiment of primary cultured SPNs *in vitro* with low concentration of ketamine (1.5 $\mu\text{g}/\text{mL}$, equal to 5.5 μM) treatment, followed by the measurement of MAGL and PRDM5 expressions by immunoblotting, as well as 2-AG detection. We found that ketamine (1.5 $\mu\text{g}/\text{mL}$) also significantly decreased MAGL expression and increased PRDM5 expression and 2-AG level in the cultured SPNs. The results were consistent with that of high concentration of ketamine (100 μM). We adopted the updating data (Fig. 7a, b, c) to replace the original one in the revised manuscript.

9. Since it is difficult to demonstrate a direct effect of ketamine on SPN neurons that would lead to increased 2-AG levels via downregulation of MAGL and up-regulation of PRDM5, what other possible mechanisms could account for these effects?

Reply: We thank the review for the comment. As reviewer pointed out, it is difficult to demonstrate a direct effect of ketamine on SPN neurons. Based on the well-known pharmacology effect of ketamine—NMDAR antagonist, we speculated that NMDAR may play an important role in ketamine-induced ECS alterations. To this end, we applied MK801 to investigate whether it would exhibit similar effect as ketamine, as both ketamine and MK801 are NMDAR antagonist. We performed several new studies as follows:

(1) We investigated the effect of MK801 on mice locomotion. Similar with ketamine's effect, MK801 obviously enhanced locomotor activity in mice. MK801 also decreased MAGL expression and increased 2-AG level in the DLS (Supplementary, Fig. S12 A-D).

(2) The cultured striatal neurons were treated with MK801. We found that MK801 decreased MAGL expression, and increased PRDM5 expression as well as 2-AG level in the cultured striatal neurons. These effects are well in line with ketamine's effects (Supplementary, Fig. S12E, F).

Above results show that MK801 exhibit similar effect as ketamine on MAGL/PRDM5 expression and 2-AG production. Thinking that ketamine and MK801 are NMDAR antagonist, we infer that the effect of ketamine may be mediated, at least in part, by NMDAR. We added these new results in the Results of the revised manuscript (Page12, Line 319~326; Page17, Line 474~479.).

REVIEWER COMMENTS

Reviewer #1 (Remarks to the Author):

The authors have provided thorough and clear, mostly satisfying revisions in response to the reviewer comments. There are only two remaining suggestions listed below to further improve the manuscript.

Original point 1: "As the reviewer suggested, we added literatures regarding the glutamatergic regulation of endocannabinoid system in the Discussion (Page 21-22, Line 593~603). Also, these literatures are listed in the Reference (references 56~59) in revised manuscript (abbreviated as revised MS)."

These are useful additions but the added text is somewhat ambiguous:

"593 In mesocorticolimbic reward circuits, the glutamatergic and endocannabinoid systems are implicated in the neurobiological mechanisms underlying drug addiction. Previous researches have demonstrated that ECS plays a central role in modulating glutamatergic function in cocaine-induced behavior [55, 56]. However, few studies focus on the glutamatergic regulation of ECS [57], especially in drug addiction."

The first and last sentences are, however, somewhat contradictory. It would be more useful to reframe in the context of the endocannabinoid regulation of the glutamate system (rather than the other way around) and synaptic function, which is an extensively studied area in the context of drug addiction. Glutamatergic regulation, both in regard to endocannabinoid regulation and drug addiction, is very well studied.

Original point 5: "No rationale is give as to why the 15 mg/kg ketamine dose was chosen for the experiments. It would be important to justify, e.g. that it is the equivalent of ketamine commonly used recreationally. Also, what was the dosing schedule in the morphological experiments presented in Fig. 3?"

Reply: We chosed the dose of 15 mg/kg for the experiments based on the following reasons:
(1) Ketamine abusers inject ketamine usually at a dose of 100~200 mg/person [1]. Assuming 60 kg per an adult, the dose of ketamine abusers is 1.67~3.33 mg/kg. Considering that biological equivalent dose coefficient (mice/human=9.1), 15~30 mg/kg in mice is almost equivalent to recreational use in human.
(2) In animal research, ketamine increases locomotor activity at the dose of 4~16 mg/kg [2]. Another animal research also showed that 30 mg/kg ketamine induces locomotion sensitization in mice [3]."

The authors have responded to the question in the Rebuttal very nicely, but the above statements have not been integrated into the MS. It would be important to include these rationalizations and references in the paper as many readers may be interested in how the dose was chosen in a broader context and the reasoning behind the animal model.

Reviewer #2 (Remarks to the Author):

The authors have done a commendable job of revising the manuscript, including new and informative experimental findings. Most of the concerns have been addressed. However, a few questions remain.

1. It is now clearer how the lipidomic analysis was performed. However, simply referring to the

web page for the analysis software does not address the original question about the handling of multiple comparisons. There is a false discovery rate analysis in the Progenesis QI software, but was this used in the lipidomic analysis in this paper? In general, it is insufficient to simply use an online analysis software package without knowing which exact analyses you are performing.

2. The response to the question about antibody specificity is likewise inadequate. The question was how the antibodies were tested for specificity, and the supplementary table does not provide this information. The links to the company websites are useful, but not always informative. For example, for the two antibodies from Santa Cruz the web page says nothing about how specificity was assessed (ideally it should be tested in knockout mice). This concern is heightened by the fact that some of the antibodies are from sources known to have specificity problems. In addition, the PDF version of supplementary table 5 lacks the column showing the website URLs, and this column is only present in the spreadsheet version.

3. Additional editing for grammar and usage is needed, including in some of the newly-added sections.

Reponse letter

Reviewer #1 (Remarks to the Author):

The authors have provided thorough and clear, mostly satisfying revisions in response to the reviewer comments. There are only two remaining suggestions listed below to further improve the manuscript.

Original point 1: "As the reviewer suggested, we added literatures regarding the glutamatergic regulation of endocannabinoid system in the Discussion (Page 21-22, Line 593~603). Also, these literatures are listed in the Reference (references 56~59) in revised manuscript (abbreviated as revised MS)."

These are useful additions but the added text is somewhat ambiguous:

"593 In mesocorticolimbic reward circuits, the glutamatergic and endocannabinoid systems are implicated in the neurobiological mechanisms underlying drug addiction. Previous researches have demonstrated that ECS plays a central role in modulating glutamatergic function in cocaine-induced behavior [55, 56]. However, few studies focus on the glutamatergic regulation of ECS [57], especially in drug addiction."

The first and last sentences are, however, somewhat contradictory. It would be more useful to reframe in the context of the endocannabinoid regulation of the glutamate system (rather than the other way around) and synaptic function, which is an extensively studied area in the context of drug addiction. Glutamatergic regulation, both in regard to endocannabinoid regulation and drug addiction, is very well studied.

Reply: We thank the review's suggestion. In this study we focused on the endocannabinoid regulation of glutamate system and synaptic function. We reframed the context in the revised MS (Page 21, Line 599).

Original point 5: "No rationale is give as to why the 15 mg/kg ketamine dose was chosen for the experiments. It would be important to justify, e.g. that it is the equivalent of ketamine commonly used recreationally. Also, what was the dosing schedule in the morphological experiments presented in Fig. 3?"

Reply: We chosed the dose of 15 mg/kg for the experiments based on the following reasons:

- (1) Ketamine abusers inject ketamine usually at a dose of 100~200 mg/person [1]. Assuming 60 kg per an adult, the dose of ketamine abusers is 1.67~3.33 mg/kg. Considering that biological equivalent dose coefficient (mice/human=9.1), 15~30 mg/kg in mice is almost equivalent to recreational use in human.
- (2) In animal research, ketamine increases locomotor activity at the dose of 4~16 mg/kg

[2]. Another animal research also showed that 30 mg/kg ketamine induces locomotion sensitization in mice [3].”

The authors have responded to the question in the Rebuttal very nicely, but the above statements have not been integrated into the MS. It would be important to include these rationalizations and references in the paper as many readers may be interested in how the dose was chosen in a broader context and the reasoning behind the animal model.

Reply: We added rationale and references regarding to the dosage of ketamine (15 mg/kg) in the revised MS (Page 5, Lines 110-111).

Reviewer #2 (Remarks to the Author):

The authors have done a commendable job of revising the manuscript, including new and informative experimental findings. Most of the concerns have been addressed. However, a few questions remain.

1. It is now clearer how the lipidomic analysis was performed. However, simply referring to the web page for the analysis software does not address the original question about the handling of multiple comparisons. There is a false discovery rate analysis in the Progenesis Q1 software, but was this used in the lipidomic analysis in this paper? In general, it is insufficient to simply use an online analysis software package without knowing which exact analyses you are performing.

Reply:

- a. Multiple comparisons arise when a statistical analysis involves multiple simultaneous statistical tests, each of which has a potential to produce a "discovery", of the same dataset or dependent datasets. As the number of comparisons increases, the probability of incorrectly rejecting the null hypothesis will also increase. Thus it need to re-calculate probabilities obtained from a statistical test which was repeated multiple times. The Bonferroni correction is one of the most commonly used approaches for multiple comparisons. However, in the large-scale testing, such as lipidomics, it is usually to use FDR (false discovery rate) to control false positive.**
- b. Progenesis Q1 software (Nonlinear Dynamics, Newcastle, UK) is a professional metabolic analysis software and has been widely used in scientific researches [1, 2]. It uses multivariate statistical analysis to find differential metabolites for untargeted metabolomics study. Q-values are the name given to the adjusted p-values found using an optimized FDR approach. The FDR approach is optimized by using characteristics of the p-value distribution to produce a list of q-values, which will result in fewer false positives. The more detailed information about q-values can be found in the reference [3].**
- c. We indeed used false discovery rate (FDR) analysis in the lipidomic analysis in our study. The original lipidomic data showed q value, which is the adjusted p-values using an optimized FDR approach. Although some lipids have large q-values, we still keep them in the initial lipidomic data to avoid missing potentially valuable lipids. After all, lipidomic is a screening tool, followed by other targeted methods. Specifically, 2-AG and AEA are our focus. Due to the high q-values in the ACe, we selected the other four brain regions (hippocampus, NAc, PFC and CPu) for the following targeted lipidomic analysis. The results showed that the alterations of 2-AG and AEA induced by ketamine in targeted lipidomic analysis were in line with the predictions of untargeted lipidomic analysis. Ketamine significantly elevated 2-AG and AEA in the CPu**

and hippocampus, in which q-values were less than 0.05 and 0.1, respectively. Thus our data also proved the validity of q-values. In addition, we added FDR analysis in Lipidomic data processing and analysis in the revised MS (Supplement, Page 3, Lines 71~72). We also supplemented the original lipidomic data in the revised MS.

2. The response to the question about antibody specificity is likewise inadequate. The question was how the antibodies were tested for specificity, and the supplementary table does not provide this information. The links to the company websites are useful, but not always informative. For example, for the two antibodies from Santa Cruz the web page says nothing about how specificity was assessed (ideally it should be tested in knockout mice). This concern is heightened by the fact that some of the antibodies are from sources known to have specificity problems. In addition, the PDF version of supplementary table 5 lacks the column showing the website URLs, and this column is only present in the spreadsheet version.

Reply:

We thank reviewer's comment. We know that it would be unpractical to test each antibody's specificity in a study. To guarantee the antibody specificity, we purchased the antibodies from the well-known and professional antibody producers which have quality control about their antibodies as much as possible. We also selected the antibodies which have been tested in other published papers. When it comes to the two PRDM5 antibodies from Santa Cruz, the antibodies are generated from human PRDM5 (Gene ID:11107), and the immunogenicity sequence is "DYRLMWEVRGSKGEVLYILDATNPRHSNWLRVHEAPSQEQKNLAAIQEGENIFYLA VEDIETDTELLIGYLDSDMEAEQQIMTVIKEGEVENSRRQSTAGRKDLGCKEDYA CPQCESS". The producer has conducted LC-MS to test the accuracy of the protein sequences of PRDM5 antibodies. Furthermore, Santa Cruz also conducted Western blotting to prove its specificity in several cell lines, and the new data can be found on their website (<https://www.scbt.com/zh/p/prdm5-antibody-a-12?requestFrom=search>). In addition, the ChIP grade PRDM5 antibody from Santa Cruz is the only commercial antibody for ChIP experiment and it has been used in the published reference [4].

We generated a new PDF version of supplementary table 5 containing the column showing the website URLs.

3. Additional editing for grammar and usage is needed, including in some of the newly-added sections.

Reply:

Thank for the suggestion and we invited some native speakers to help us editing the manuscript.

Response references:

1. Li, H., et al., *Metabolomic adaptations and correlates of survival to immune checkpoint blockade*. Nature communications, 2019. **10**(1): p. 4346.
2. Luengo, A., et al., *Reactive metabolite production is a targetable liability of glycolytic metabolism in lung cancer*. Nature communications, 2019. **10**(1): p. 5604.
3. Storey, J.D. and R. Tibshirani, *Statistical significance for genomewide studies*. Proc Natl Acad Sci U S A, 2003. **100**(16): p. 9440-5.
4. Galli, G.G., et al., *Prdm5 suppresses Apc(Min)-driven intestinal adenomas and regulates monoacylglycerol lipase expression*. Oncogene, 2014. **33**(25): p. 3342-50.

REVIEWERS' COMMENTS

Reviewer #2 (Remarks to the Author):

All remaining concerns have been addressed adequately.

Reponse letter

Reviewer #2 (Remarks to the Author):

All remaining concerns have been addressed adequately.

Reply:

Thank you very much.